# Rehydration rescues *Il22*−/− mice from lethal *Citrobacter rodentium* infection

**Vishwas Mishra** [1] ✉, **Zuza Kozik**[2,3], **Priyanka Biswas** [1,3], **Jyoti Choudhary** [2], **Joshua L. C. Wong** [1] **& Gad Frankel** [1] ✉

Interleukin-22 (IL-22) is considered indispensable for host defence against *Citrobacter rodentium*, with 100% mortality of *Il22*−/− mice. While IL-22 promotes epithelial barrier integrity and production of antimicrobial peptides, the precise mechanism underlying lethality remains unclear. Here, we show that following *C. rodentium* infection *Il22*−/− mice succumb due to dehydration, rather than failure to control bacterial burden or regenerate damaged intestinal epithelium. Proteomic and gene expression analysis reveal greater enterocyte depletion in *C. rodentium*-infected *Il22*−/− mice, resulting in significant reductions in ion transporter abundances. We show that while not reducing bacterial load, improving the gut barrier integrity, or affecting immune responses, fluid therapy (FT) fully rescues *Il22*−/− mice by correcting systemic dehydration. Survival is associated with locally increased *Reg3b*, IL-17F and IL-10 levels, suggesting activation of compensatory pathways that potentially support recovery in the absence of IL-22. Recovered *Il22*−/− mice exhibit epithelial cell regeneration and tissue physiology similarly to *C. rodentium*-infected *Il22*+/+ mice. These findings suggest that dehydration is the primary cause of mortality in *Il22*−/− mice and reveal that IL-22 prevent this outcome by preserving epithelial integrity and fluid-ion absorption. Importantly, this study underscores the necessity of incorporating supportive therapies into preclinical infection models to better reflect physiological settings and improve their relevance in modelling human disease.

*Citrobacter rodentium* is the aetiological agent of transmissible murine colonic crypt hyperplasia[1,2]. *C. rodentium* is an extracellular Gram-negative mouse-specific pathogen; it causes self-limiting infection in C57BL/6 mice, which develop colitis and mild diarrhoea[3]. Infection of mice with *C. rodentium* serves as a robust preclinical model for studying enteric infections (e.g. enteropathogenic *Escherichia coli* (EPEC) and enterohaemorrhagic *E. coli* (EHEC)), colitis and gut recovery following epithelial damage[3–5]. *C. rodentium* targets absorptive enterocytes, attaching intimately to their apical surface and disrupting intestinal homoeostasis and reshaping the intestinal cellular composition[6,7]. Infection is mediated by injection of effectors (e.g. Tir, EspF) via a type III secretion system, which subvert cell signalling in

intestinal epithelial cells (IECs)[7,8]. *C. rodentium* infection triggers loss of mucin-producing goblet cells and absorptive enterocytes, coupled with an expansion of transit-amplifying cells, culminating in colonic crypt hyperplasia (CCH)[2,7]. In addition, *C. rodentium* disrupts tight junctions and the intestinal barrier integrity, triggering robust inflammatory responses[1,2,5]. Host defence mechanisms against *C. rodentium* rely on type 3 immunity, characterised by production of interleukin (IL)−17 and IL-22, which plays a critical role in mucosal immunity[6,9–11].

Perturbation of the gut barrier, either chemically via dextran sulphate sodium (DSS) or by *C. rodentium*, *Salmonella Typhimurium*, or *Clostridioides difficile* infection, triggers the secretion of IL-22[12–14]. IL-22

[1]Department of Life Sciences, Imperial, London SW7 2AZ, UK. [2]Functional Proteomics Group, Chester Beatty Laboratories, Institute of Cancer Research, London, UK. [3]These authors contributed equally: Zuza Kozik, Priyanka Biswas. ✉e-mail: v.mishra@imperial.ac.uk; g.frankel@imperial.ac.uk

is a member of the IL-10 family of cytokines and is primarily produced by group 3 innate lymphoid cells (ILC3s) and Th17/Th22 cells[11,13,15]. ILC3s serve as the primary source of IL-22 during the early phase of *C. rodentium* infection where it targets superficial IECs[10]. Recent findings showed that a brief ciprofloxacin treatment administered 4 days post infection (dpi) induces persistent ILC3 activation[16]. These "trained" ILC3s exhibit an enhanced capacity to produce IL-22, conferring greater protection against subsequent infections[11,16]. As the infection burden increases and *C. rodentium* colonises a large surface of the colonic mucosa, IL-22 production shifts from ILC3s to CD4⁺ T cells. CD4⁺ T cell-derived IL-22 plays a crucial role in preventing bacterial invasion of colonic crypts and limiting systemic dissemination[6,10].

IL-22 exerts its effects on IECs through binding to the IL-22 receptor, composed of an IL-22R1 and IL-10R2 heterodimer[13,17]. Binding to the IL-22R triggers activation of the Jak kinase Stat transcription factor pathway, predominantly Jak1, Tyk2 and STAT3[11,13,18]. Upon phosphorylation, STAT3 dimers translocate to the nucleus where they function as transcriptional activators, regulating expression of antimicrobial proteins (AMPs) and multiple cellular functions, including chemotaxis, proliferation, acute-phase responses, innate immunity, inflammation and tissue healing[13,15]. Amongst the STAT3 regulated proteins are calprotectin (S100A8/S100A9 heterodimer)[19], a metal (Mn- and Zn)-sequestering complex, LCN-2[20], which inhibits bacterial growth by sequestering the siderophore enterobactin, and the Reg family (Reg3β/γ) of AMPs[21]. The IL-22-STAT3 axis enhances epithelial barrier integrity by upregulating tight junction proteins (e.g., claudins and occludins) and mucins (e.g., MUC1 and MUC2)[13,15]. Expression of claudins, calprotectin, LCN-2 and Reg3γ are detected as early as 2- to 4-dpi[22,23].

A seminal publication by Zheng et al. showed that IL-22 is vital for protecting mice against *C. rodentium* infection, where 100% mortality of *Il22⁻/⁻* mice was recorded[9]. Since then, there has been a concerted scientific effort aimed at determining the mechanism underpinning this phenotype[6,10,24–28]. Germ-free *Il22⁻/⁻* mice succumb to *C. rodentium* infection similarly to specific pathogen free (SPF) *Il22⁻/⁻* mice, suggesting that mortality is independent of gut microbiota[27]. Moreover, *Stat3*-deficient mice also succumb to *C. rodentium* infection[29]. However, mice deficient in several IL-22/STAT3 regulated proteins individually survive and clear the infection[28]. While *Il22⁻/⁻* mice survive infection with *C. rodentium-ΔespF* (EspF is an effector that disrupts TJ)[30], the precise cause of death of *C. rodentium*-infected *Il22⁻/⁻* mice remains unknown. Notably, *C. rodentium* infection induces severe diarrhoeal symptoms in *Il22⁻/⁻* mice[9,27,30], raising the possibility that dehydration may be the primary driver of mortality.

This study aimed to determine the cause of death of *C. rodentium*-infected *Il22⁻/⁻* mice. We demonstrate that dehydration, rather than uncontrolled bacterial burden or inability to restore the gut barrier functions, is the primary cause of fatality. Furthermore, we show that fluid therapy (FT) alone is sufficient to rescue *Il22⁻/⁻* mice, suggesting the existence of IL-22-independent mechanisms that can take over bacterial clearance and tissue repair functions when hydration status is restored.

## Results

### C. rodentium infection causes severe diarrhoea in Il22⁻/⁻ mice

We infected C57BL/6 *Il22⁻/⁻* mice with *C. rodentium* by oral gavage, infection of *Il22⁺/⁺* mice was used as a control. As reported earlier[9,10,25,27,28,30] (Table S1), *Il22⁻/⁻* mice exhibited significant temporal weight loss (Fig. 1A) and reached 100% mortality by 14 dpi (Fig. 1B). Prior to reaching the endpoint, the *Il22⁻/⁻* mice developed severe diarrhoea, characterised by visibly loose stool and faecal matter adhering to the cage walls (Fig. 1C). To quantify diarrhoea severity, we measured faecal trait score using the modified Bristol stool scale adapted for laboratory mice[31] and determined faecal water content (Fig. 1D, E). Both *Il22⁺/⁺* and *Il22⁻/⁻* mice showed increased faecal trait

scores and water content upon *C. rodentium* infection, indicating diarrhoea, but values were significantly higher in *Il22⁻/⁻* mice at 6 dpi (Fig. 1D, E). In addition, only infected *Il22⁻/⁻* mice exhibited faecal sodium and potassium ion losses, indicating electrolyte imbalance as a contributor to disease severity (Fig. 1F, G).

Given the diarrhoeal phenotype, we next investigated whether underlying epithelial changes, particularly in ion transport and fluid regulation could explain the electrolyte imbalance observed in *Il22⁻/⁻* mice. As an exploratory step, we performed deep quantitative proteomics analysis on colonic IECs isolated from *C. rodentium* infected *Il22⁺/⁺* and *Il22⁻/⁻* mice at 9 dpi, using two biological repeats per condition. Among the 7730 quantified proteins, 690 were differentially expressed between *Il22⁺/⁺* and *Il22⁻/⁻* mice post-infection (p < 0.05, |log2FC| > 0.5) (Fig. S1A). While the small sample limited the statistical power and precluded FDR correction, gene set enrichment analysis of these differentially regulated proteins revealed significant down-regulation of pathways associated with ion transport (Fig. S1B). Upon *C. rodentium* infection *Il22⁻/⁻* mice exhibited a notable depletion of key ion transporters and regulators involved in electrolyte and water absorption in the colon (Fig. 2A, Fig. S1C). We validated the down-regulation of ion transporters implicated in electrolyte and water absorption[32] by qRT-PCR. Gene expression levels of the IL-22-induced AMPs *Reg3b* and *Reg3g* were used as controls. *Il22⁻/⁻* mice displayed lower expression of *Reg3b* and *Reg3g* both at baseline and post-infection compared to *Il22⁺/⁺* mice (Fig. 2B). As previously reported[13], these AMPs were upregulated in *C. rodentium* infected *Il22⁺/⁺* mice; notably, they were also upregulated in *Il22⁻/⁻* mice, indicating IL-22-independent regulation (Fig. 2B). The abundance and expression of ion transporters, including Chloride anion exchanger (*Slc26a3*), a chloride-bicarbonate exchanger essential for chloride absorption, Carbonic anhydrases 2 (*Ca2*) and 4 (*Ca4*) involved in bicarbonate absorption, Aquaporin-8 (*Aqp8*) which facilitates water absorption, *Slc5a8* which functions as a sodium-coupled transporter for short-chain fatty acids, D-lactate, and monocarboxylates and *Slc15a1* (*Pept1*) a proton-dependent peptide transporter that absorbs di-/tripeptides, was significantly reduced in both *C. rodentium*-infected *Il22⁺/⁺* and *Il22⁻/⁻* mice, except for *Slc5a8*, which was downregulated only in *Il22⁻/⁻* mice (Fig. 2C). Importantly, expression of all these ion transporters was significantly lower in infected *Il22⁻/⁻* mice compared to *Il22⁺/⁺* mice (Fig. 2C). This further reduction provides a mechanistic basis for the severe diarrhoea phenotype observed in infected *Il22⁻/⁻* mice.

Since transporter downregulation appeared central to the diarrheal phenotype, we next asked whether IL-22 directly controls their expression in colonic epithelial cells. To this end, we first analysed publicly available gene expression data from mouse colonic organoids treated with recombinant IL-22 (rIL-22)[33] (Fig. S2A, Table S2). As expected, rIL-22 treatment upregulated established IL-22-induced genes, including *Reg3b*, *Reg3g*, *S100a8*, *Cxcl5*, and *Retnlb*; however, *Slc26a3*, *Ca2*, *Slc5a8*, and *Aqp8* expression was downregulated in rIL-22-treated organoids (Fig. S2A, Table S2). To assess regulation in vivo, we next treated naïve mice with rIL-22, and analysed colonic gene expression. rIL-22-treated mice showed increased *Reg3b* and *Reg3g* expression, confirming effective IL-22 activity, but exhibited no significant changes in *Slc26a3*, *Ca2*, *Ca4*, *Aqp8*, *Slc5a8*, or *Slc15a1* (Fig. S2B-C). These findings indicate that IL-22 does not directly regulate transcription of these ion transporters in the mouse colon.

Since transporter expression was not directly induced by IL-22, we investigated whether the loss was due to infection-induced changes in IEC subpopulations. Using publicly available single-cell RNA-sequencing data from mouse colon[34] (Single Cell Portal, accession SCP1891), we examined ion channel expression across epithelial subtypes (Fig. 2D). We found that expression of *Slc26a3*, *Ca4*, *Aqp8*, and *Slc15a1* is restricted to the enterocytes, consistent with *Slc26a3* and *Ca4* being established enterocyte markers[6]. While *Ca2* and *Slc5a8* are expressed in stem cells, the highest expression is seen in enterocytes (Fig. 2D). To

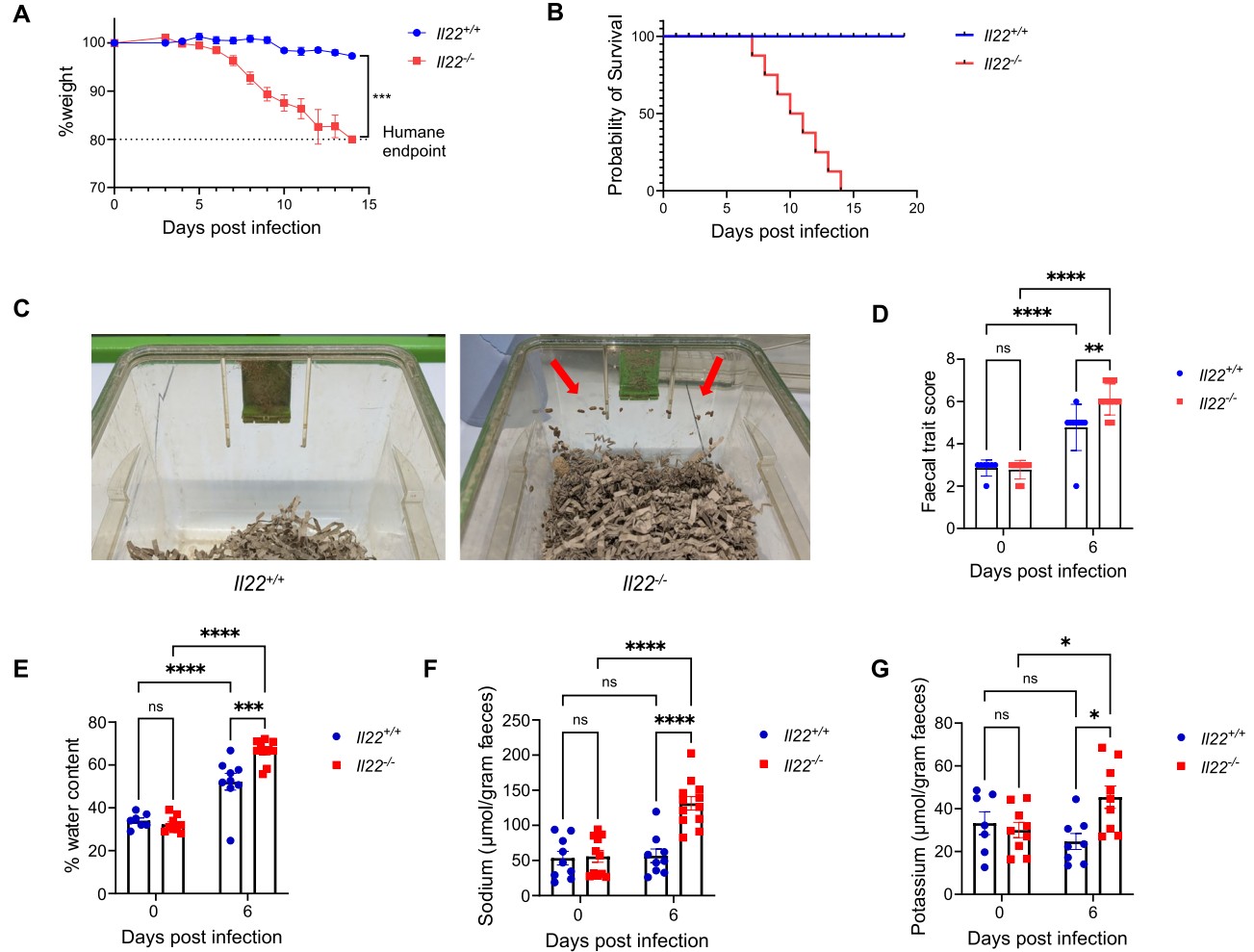

**Fig. 1 | *C. rodentium*-infected *Il22*⁻/⁻ mice develop severe diarrhoea with electrolyte imbalance. A** Temporal weight loss of *C. rodentium*-infected *Il22*⁺/⁺ and *Il22*⁻/⁻ mice. **B** Probability of survival. **C** Representative images of cages at 6 dpi; note loose stools adhering to cage walls in *Il22*⁻/⁻ mice. **D, E** Faecal trait scores determined by the modified Bristol stool scale (**D**) and faecal water content (**E**) at 0 and 6 dpi. **F, G** Faecal sodium (**F**) and potassium (**G**) levels at 0 and 6 dpi. Data in **A**, **B** are pooled values from 5 biological repeats, with 3-5 mice per group per repeat. Each dot in (**D–G**) represents an individual mouse, with pooled values from 2 biological repeats (4-5 mice per group). Refer to Supplementary Data 1 for exact group sizes. P values were determined on data plotted as Mean ± SEM using two-tailed Student's t-test in (**A**) and Two-way ANOVA with Bonferroni post-test for multiple comparisons in (**D–G**). ns, non-significant; *, $p < 0.05$, **, $p < 0.01$; ***, $p < 0.001$; ****, $p < 0.0001$. Source data with all raw values and exact p values are provided as a Source Data file.

assess infection-driven changes in IEC composition, we performed qRT-PCR for cell type-specific marker genes (Fig. 2E). Stem cell marker *Lgr5* and enteroendocrine cell marker *Chga* were unchanged following *C. rodentium* infection. In contrast, tuft cell marker *Dclk1* and goblet cell marker *Muc2* were significantly downregulated in both *Il22*⁺/⁺ and *Il22*⁻/⁻ mice. Enterocyte markers *Alpi* and *Krt20* were reduced in both genotypes, with significantly greater downregulation in infected *Il22*⁻/⁻ mice compared to *Il22*⁺/⁺ controls. These findings suggest that increased epithelial damage and a greater loss of the enterocyte population may contribute to the severe diarrhoea observed in *Il22*⁻/⁻ mice.

### Dehydration drives mortality in *Il22*⁻/⁻ mice following *C. rodentium* infection

Since *Il22*⁻/⁻ mice exhibited diarrhoea and significant depletion of key ion transporters, we hypothesised that dehydration was the primary cause of mortality. To test this hypothesis, we implemented a fluid therapy (FT) regimen aimed at restoring hydration status and assessing its impact on survival (Fig. 3A). Starting at 5 dpi, coinciding with the onset of diarrhoea, infected *Il22*⁻/⁻ mice received daily subcutaneous injections of a balanced salt solution until 20 dpi (Fig. 3A,

see methods and Table S3 for details). While treated mice continued to present diarrhoea (Fig. 3B), FT successfully normalised serum dehydration parameters[35] in *Il22*⁻/⁻ mice. Serum protein concentration increased significantly in infected *Il22*⁻/⁻ compared to *Il22*⁺/⁺ control mice (Fig. 3C), consistent with dehydration. However, serum protein concentration was normalised in *C. rodentium*-infected *Il22*⁻/⁻ mice with FT (Fig. 3C). Both *Il22*⁻/⁻ and *Il22*⁺/⁺ mice showed elevated serum renin and corticosterone following infection; however, increases were greater in *Il22*⁻/⁻ mice, and FT resulted in significantly lower levels compared to untreated mice (Fig. 3D-E).

Acute renal failure is an established consequence of dehydration[36,37]. To assess renal function, we measured serum cystatin C, which was significantly elevated in infected *Il22*⁻/⁻ mice compared to *Il22*⁺/⁺ mice; the levels of cystatin C were restored by FT (Fig. 3F). Clinical assessment further revealed substantial improvement in hydration status in FT-treated *Il22*⁻/⁻ mice, as reflected by disease scores showing reductions in ruffled coat appearance, increased skin turgor, improved posture, and enhanced mobility compared to untreated *Il22*⁻/⁻ mice (Fig. 3G). These findings indicate that FT effectively reverses the dehydration parameters observed in infected *Il22*⁻/⁻ mice. Importantly, administration of FT resulted in complete survival

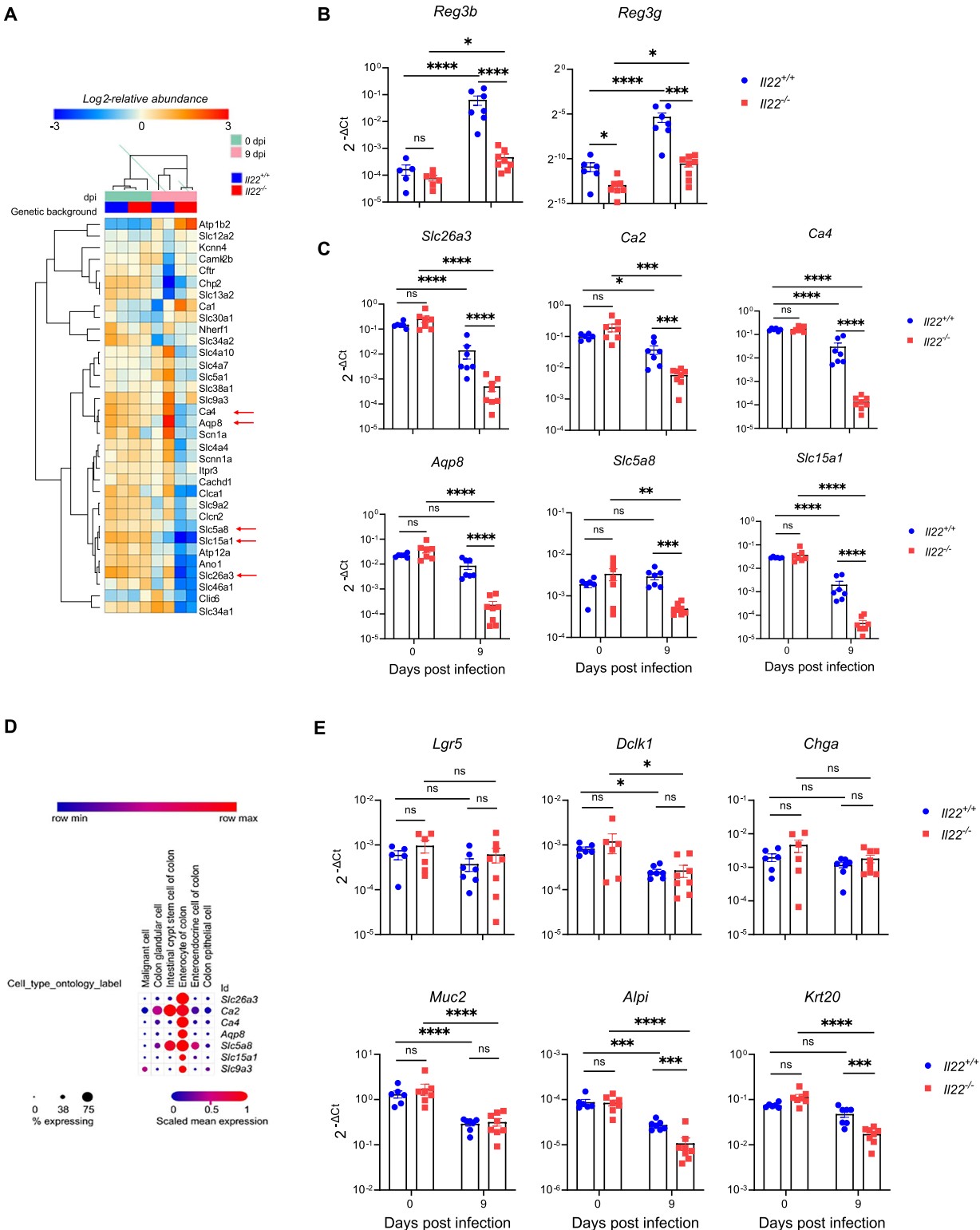

**Fig. 2 | Loss of absorptive enterocytes underlies ion transporter depletion in *C. rodentium*-infected *Il22*^−/− mice.** **A** Proteomics heat-map of colonic IEC transporters and regulators involved in fluid-ion homoeostasis at 0 and 9 dpi in *Il22*^+/+ and *Il22*^−/− mice. **B**, **C** qRT-PCR of *Reg3b* and *Reg3g* (**B**) and colonic ion transporters (*Slc26a3*, *Ca2*, *Ca4*, *Aqp8*, *Slc5a8*, *Slc15a1*) (**C**) at 0 and 9 dpi. **D** Expression of selected ion transporters across intestinal epithelial cell subtypes analysed on Single cell Portal based on publicly available scRNA-seq data (SCP1891). **E** qRT-PCR of epithelial subtype markers: *Lgr5* (stem cells), *Dclk1* (tuft), *Chga* (enteroendocrine), *Muc2* (goblet), *Alpi* and *Krt20* (enterocytes) at 0 and 9 dpi. Each dot represents an individual mouse. Data shown are pooled values from 2 biological repeats with 3-5 mice per group. Refer to Supplementary Data 1 for exact sample sizes. Data are shown as Mean ± SEM. P values determined by Two-way ANOVA on log2 values with Bonferroni post-test. ns, non-significant; *, $p < 0.05$; **, $p < 0.01$; ***, $p < 0.001$; ****, $p < 0.0001$. Source data with all raw values and exact p values are provided as a Source Data file.

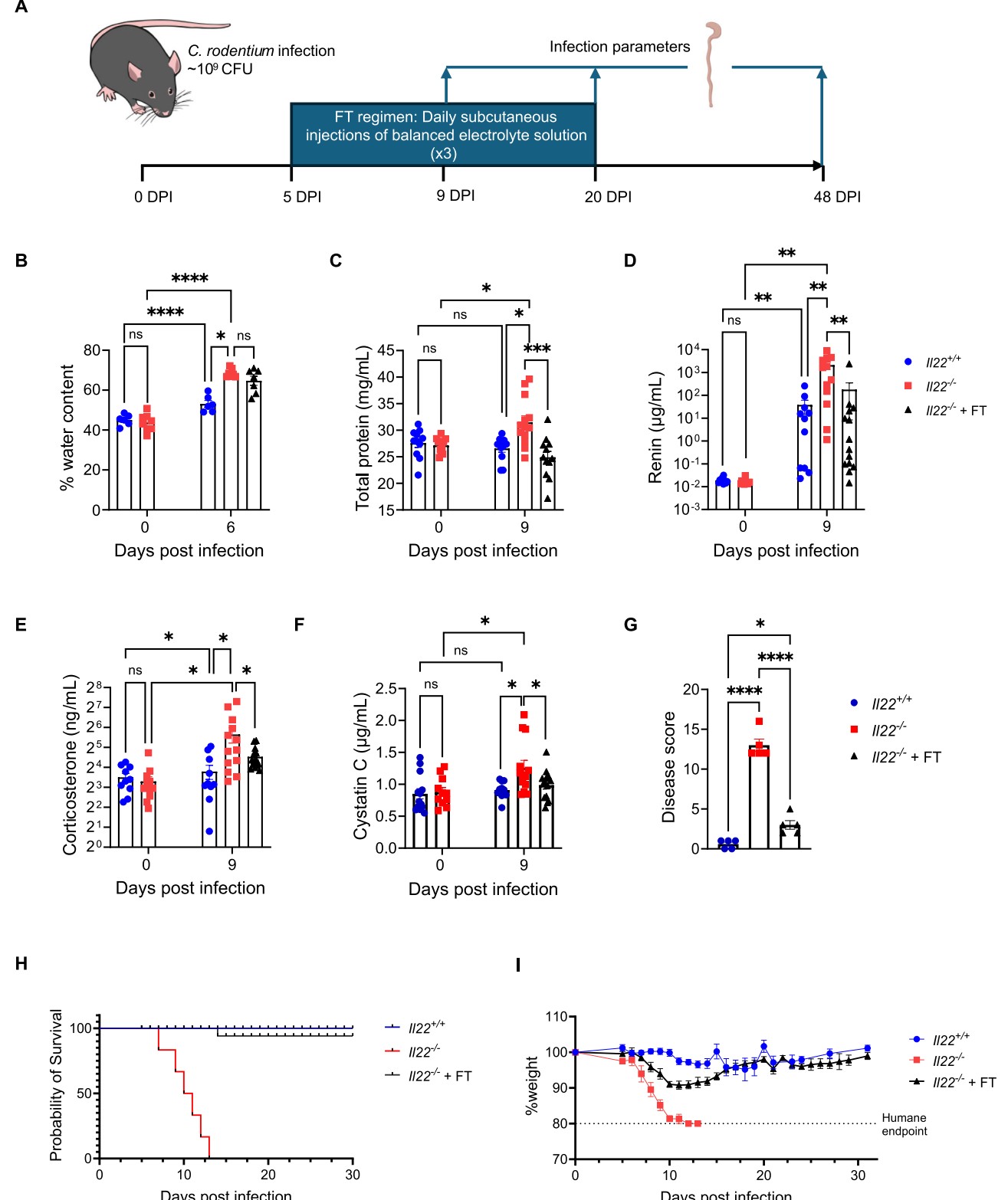

**Fig. 3 | FT rescues *C. rodentium*-infected *Il22⁻ᐟ⁻* mice from dehydration and mortality. A** Experimental schematic of *C. rodentium* infection and FT regimen. **B** Faecal water content at 0 and 6 dpi. **C–F** Serum total protein concentration (**C**), renin (**D**), corticosterone (**E**), and cystatin C levels (**F**) in *Il22⁺ᐟ⁺*, *Il22⁻ᐟ⁻* mice ± FT at 0 and 9 dpi. **G** Clinical disease scores assessing coat, posture, mobility, and skin turgor. **H** Probability of survival. **I** Temporal body weight changes in *C. rodentium*-infected *Il22⁺ᐟ⁺*, *Il22⁻ᐟ⁻* mice ± FT. Each dot represents an individual mouse in (**B–G**). Data shown are pooled values from 2 biological repeats (**B**, **G**) and 3 biological repeats (**C–F**) with 3-5 mice per group. Data in (**H–I**) are pooled values from 3 biological repeats, with 3-5 mice per group per repeat. Refer to Supplementary Data 1 for exact group sizes. Data shown as Mean ± SEM. Statistical analyses performed by Two-way ANOVA with Bonferroni post-test (**B–G**). ns, non-significant; *, p < 0.05; **, p < 0.01; ***, p < 0.001; ****, p < 0.0001. Source data with all raw values and exact p values are provided as a Source Data file. The mouse cartoon in (**A**) is an Illustration from NIAID NIH BioArt Source (bioart.niaid.nih.gov/bioart/281).

of *C. rodentium*-infected *Il22*⁻/⁻ mice, indicating that dehydration is the primary driver of mortality (Fig. 3H). While FT did not prevent weight loss during the early phase of infection, FT-treated *Il22*⁻/⁻ mice began to recover from 9 dpi, ultimately regaining body weight comparable to that of *Il22*⁺/⁺ mice by 20 dpi (Fig. 3I).

### FT does not mitigate *C. rodentium*-mediated pathology in infected *Il22*⁻/⁻ mice

While FT prevented mortality in *C. rodentium*-infected *Il22*⁻/⁻ mice, it remained unclear whether it also reduced bacterial load or improved infection-associated disease parameters and tissue pathology. Infected *Il22*⁻/⁻ mice exhibited higher faecal *C. rodentium* loads during the early (5 dpi) and peak (7-9 dpi) phases of infection compared to *Il22*⁺/⁺ mice, with no difference between FT-treated and untreated mice (Fig. 4A). As *C. rodentium* is known to colonise deep colonic crypts in *Il22*⁻/⁻ mice[9,10], we examined *C. rodentium* localisation by staining colonic sections and found no difference between FT-treated and untreated *Il22*⁻/⁻ mice, indicating that FT does not alter the colonic *C. rodentium* colonisation sites (Fig. 4B). Necropsy at 9 dpi showed that FT did not reduce systemic bacterial dissemination, with comparable *C. rodentium* counts detected in the liver and spleen in both FT-treated and untreated *Il22*⁻/⁻ mice, indicative of similar intestinal barrier disruption (Fig. 4C). Indeed, intestinal permeability measured by FITC–dextran assay (Fig. 4D), and epithelial barrier integrity assessment by E-cadherin immunohistochemistry (Fig. 5F, Fig. S3A) revealed significant impairment in infected *Il22*⁻/⁻ mice, with no improvement following FT.

Deep quantitative proteomic analysis of colonic IECs from infected *Il22*⁺/⁺ and *Il22*⁻/⁻ mice, with or without FT, compared to uninfected mice revealed no global differences in protein expression between FT-treated and untreated *Il22*⁻/⁻ mice (Fig. 4E). Given the proteomics limitations, key findings were validated using independent approaches. In agreement with the proteomic data, FT did not alter the expression of colonic ion transporters or epithelial-subset marker genes, as confirmed by qRT-PCR (Fig. S4). Altogether, these findings suggest that FT does not alter *C. rodentium* burden, localisation, or the epithelial proteome but instead functions as a supportive measure to counteract dehydration.

Further post-mortem analysis confirmed that FT did not alleviate *C. rodentium*-induced colonic damage in *Il22*⁻/⁻ mice at 9 dpi. Significant colon shortening and an increased colon weight-to-length ratio were observed in both *Il22*⁻/⁻ and *Il22*⁺/⁺ mice following *C. rodentium* infection, consistent with infectious colitis[38] (Fig. 4F-H). However, compared to infected *Il22*⁺/⁺ controls, *C. rodentium*-infected *Il22*⁻/⁻ mice, regardless of FT administration, exhibited significantly greater colon shortening (Fig. 4F-G) and a higher colon weight-to-length ratio (Fig. 4H), indicative of heightened inflammation and epithelial erosion[38]. Histological examination by haematoxylin and eosin (H&E) staining revealed comparable epithelial damage, crypt loss, mucus layer depletion, mucosal thickening, and immune cell infiltration in FT-treated and untreated *Il22*⁻/⁻ mice (Fig. 4I-J). Despite the greater epithelial damage and inflammation, CCH was similar between infected *Il22*⁺/⁺ and *Il22*⁻/⁻ mice (Fig. 4K). However, the proliferative epithelial zone, assessed by PCNA staining, was higher in *Il22*⁻/⁻ mice with or without FT (Fig. 4L, S5), suggesting a possible underestimation of CCH in *Il22*⁻/⁻ mice at 9 dpi due to extensive epithelial erosion. Together, these findings demonstrate that while FT effectively counteracts dehydration, it does not reduce the pathological effects of *C. rodentium* infection in the colon of *Il22*⁻/⁻ mice.

### *C. rodentium*-infected *Il22*⁻/⁻ mice present altered neutrophil dynamics

Since FT corrected dehydration without reducing colonic pathology, we next investigated whether systemic immune cell dynamics were altered. To investigate this, we assessed haematopoietic and immune cell changes by performing complete blood counts (CBC) on *Il22*⁺/⁺ and

*Il22*⁻/⁻ mice before and after infection. *C. rodentium* infection did not affect circulating red blood cell counts; however, both *Il22*⁺/⁺ and *Il22*⁻/⁻ mice exhibited significant reductions in white blood cells, lymphocytes, and platelets (Fig. 5A–D). The magnitude of white blood cell and platelet reduction was comparable between genotypes, whereas lymphocyte levels were significantly lower in *Il22*⁻/⁻ mice, with or without FT, compared to *Il22*⁺/⁺ mice (Fig. 5B–D). Notably, circulating neutrophil levels were unchanged in infected *Il22*⁺/⁺ mice but significantly elevated in *Il22*⁻/⁻ mice; this increase was partially reduced by FT (Fig. 5E).

To determine whether elevated circulating neutrophils corresponded to changes in the colon, we performed Ly6G immunohistochemistry analysis on colonic sections (Fig. 5F, S3A). No neutrophil staining was detected in uninfected *Il22*⁺/⁺ and *Il22*⁻/⁻ mice, whereas infection resulted in prominent neutrophil infiltration in the submucosa and mucosa. Consistent with circulating cell counts and recent reports[28], *Il22*⁻/⁻ mice displayed increased submucosal neutrophil staining compared to *Il22*⁺/⁺ mice, with no reduction following FT (Fig. 5F, S3A). Despite higher submucosal neutrophil abundance, neutrophil elastase and myeloperoxidase levels in colonic explant cultures did not differ significantly between infected *Il22*⁺/⁺ and *Il22*⁻/⁻ mice (Fig. 5G, H). Importantly, in line with the known role of IL-22 in recruiting neutrophils to the colonic epithelium in mice[10] and colitis patients[39], *Il22*⁻/⁻ mice exhibited lower proportions of mucosal neutrophils despite increased submucosal accumulation (Fig. S3B).

### FT alters cytokine levels but not immune cell composition in *Il22*⁻/⁻ mice

To further evaluate the impact of FT on local immune cell populations and inflammatory responses, we performed flow cytometry of colonic immune cells (Fig. S6A) and measured cytokine levels in the colonic explant cultures (Fig. 6). *C. rodentium* infection significantly increased colonic T, CD4⁺ T, Th17, and Th1 cells, macrophages, and Ly6C⁺ monocytes in both *Il22*⁺/⁺ and *Il22*⁻/⁻ mice, with no genotype differences or effect of FT (Fig. S6B–G). Treg and Th2 cell frequencies remained unchanged (Fig. S6H-I). In line with Ly6G immunostaining, neutrophil abundance increased upon infection, with *Il22*⁻/⁻ mice displaying significantly higher colonic neutrophil numbers than *Il22*⁺/⁺ mice at 9 dpi; FT did not impact this increase (Fig. S6J).

As expected, cytokine analysis in colonic explant cultures revealed no detectable IL-22 in *Il22*⁻/⁻ mice (Fig. 6A). Both *C. rodentium*-infected *Il22*⁺/⁺ and *Il22*⁻/⁻ mice showed elevated IFNγ, TNF, IL-17A, and IL-17F (Fig. 6B, C, E, F). Consistent with the known antagonism of IFNγ signalling by IL-22[40] and previous reports of enhanced IFNγ-driven pathways in *C. rodentium*-infected *Il22*⁻/⁻ mice[6,10], both FT-treated and untreated *C. rodentium*-infected *Il22*⁻/⁻ mice displayed significantly higher colonic IFNγ and TNF than *Il22*⁺/⁺ mice (Fig. 6B-C). Consistently, FT did not alter IFNγ-regulated protein abundances in IECs (Fig. 6D). IL-17A and IL-6 levels were comparable between *Il22*⁺/⁺ and *Il22*⁻/⁻ mice and unaffected by FT (Fig. 6E, H). In contrast, FT-treated *Il22*⁻/⁻ mice had significantly higher IL-17F and IL-10 compared to untreated *Il22*⁻/⁻ control mice (Fig. 6F, G). Given that IL-17 overlaps functionally with IL-22 and can induce *Reg3b* and *Reg3g*[41,42], we analysed their gene expression. While no differences in the levels of *Reg3b* were seen, FT treated *Il22*⁻/⁻ mice showed significantly higher expression of *Reg3g* compared to untreated *Il22*⁻/⁻ mice (Fig. S7A, B), suggesting activation of alternative recovery processes when hydration is restored.

### FT enables long term recovery in *Il22*⁻/⁻ mice

Despite persistent *C. rodentium*-associated pathology and higher faecal burden in the early and peak infection phases (Fig. 4A), FT-treated *Il22*⁻/⁻ mice cleared *C. rodentium* at rates comparable to *Il22*⁺/⁺ mice (Fig. 7A) and progressively recovered from colonic damage. No detectable CFUs were found in the liver or spleen at 20 or 48 dpi. Colon length in FT-treated *Il22*⁻/⁻ mice increased by 20 dpi compared to 9 dpi,

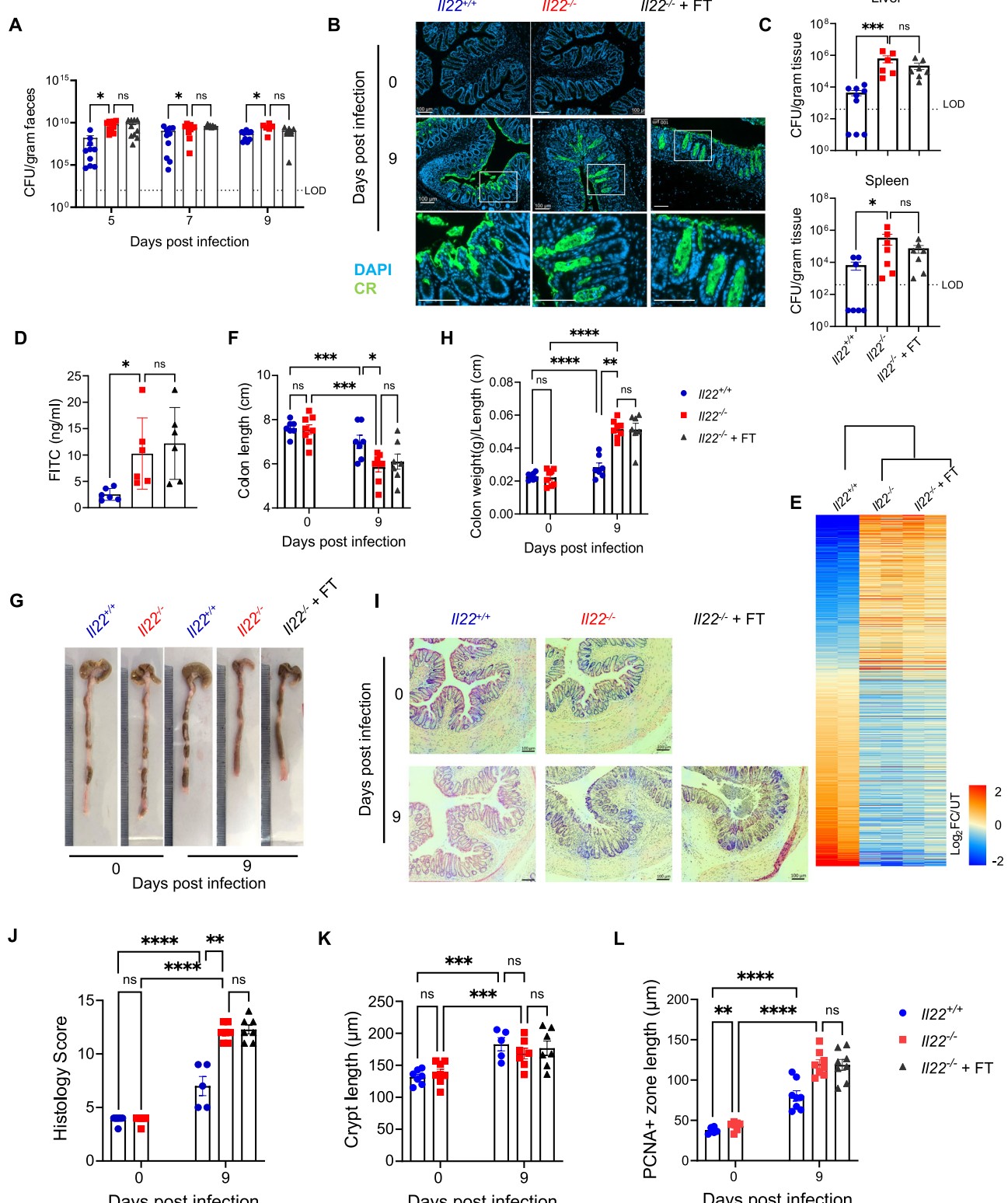

**Fig. 4 | FT does not mitigate *C. rodentium*-mediated pathology in *Il22*⁻/⁻ mice.**
**A** Faecal *C. rodentium* CFUs at 5, 7 and 9 dpi. **B** Representative distal colon sections stained for *C. rodentium* (green); note the deep crypt localisation of *C. rodentium* in *Il22*⁻/⁻ mice. DAPI was used to stain nuclei. Scale bar: 100 μm. **C** *C. rodentium* burden in liver and spleen at 9 dpi. **D** Serum FITC–dextran levels to assess intestinal permeability. **E** Heat map for quantitative proteomics analysis of colonic IECs from *Il22*⁺/⁺ and *Il22*⁻/⁻ mice ± FT at 9 dpi. **F–H** Colon length (**F**) and representative colon images (**G**) and colon weight-to-length ratio (**H**) at 0 and 9 dpi. **I** Representative images of H&E-stained distal colon sections. Scale bar: 100 μm. **J, K** Histopathology

scores (**J**) and CCH measurements (**K**) at 0 and 9 dpi. **L** Measurements of the colonic crypt PCNA⁺ zone from IHC analysis of PCNA as shown in Fig. S4. Each dot represents an individual mouse. Data shown are pooled values from 2 biological repeats with 3-5 mice per group. Refer to Supplementary Data 1 for exact group sizes. Data plotted as Mean ± SEM. P values determined by Two-way ANOVA in (**A**, **F**, **H**, **J**–**L**) and One-way ANOVA in (**C**, **D**) with Bonferroni post-test. ns, non-significant; *, $p < 0.05$; **, $p < 0.01$; ***, $p < 0.001$; ****, $p < 0.0001$. Source data with all raw values and exact p values are provided as a Source Data file.

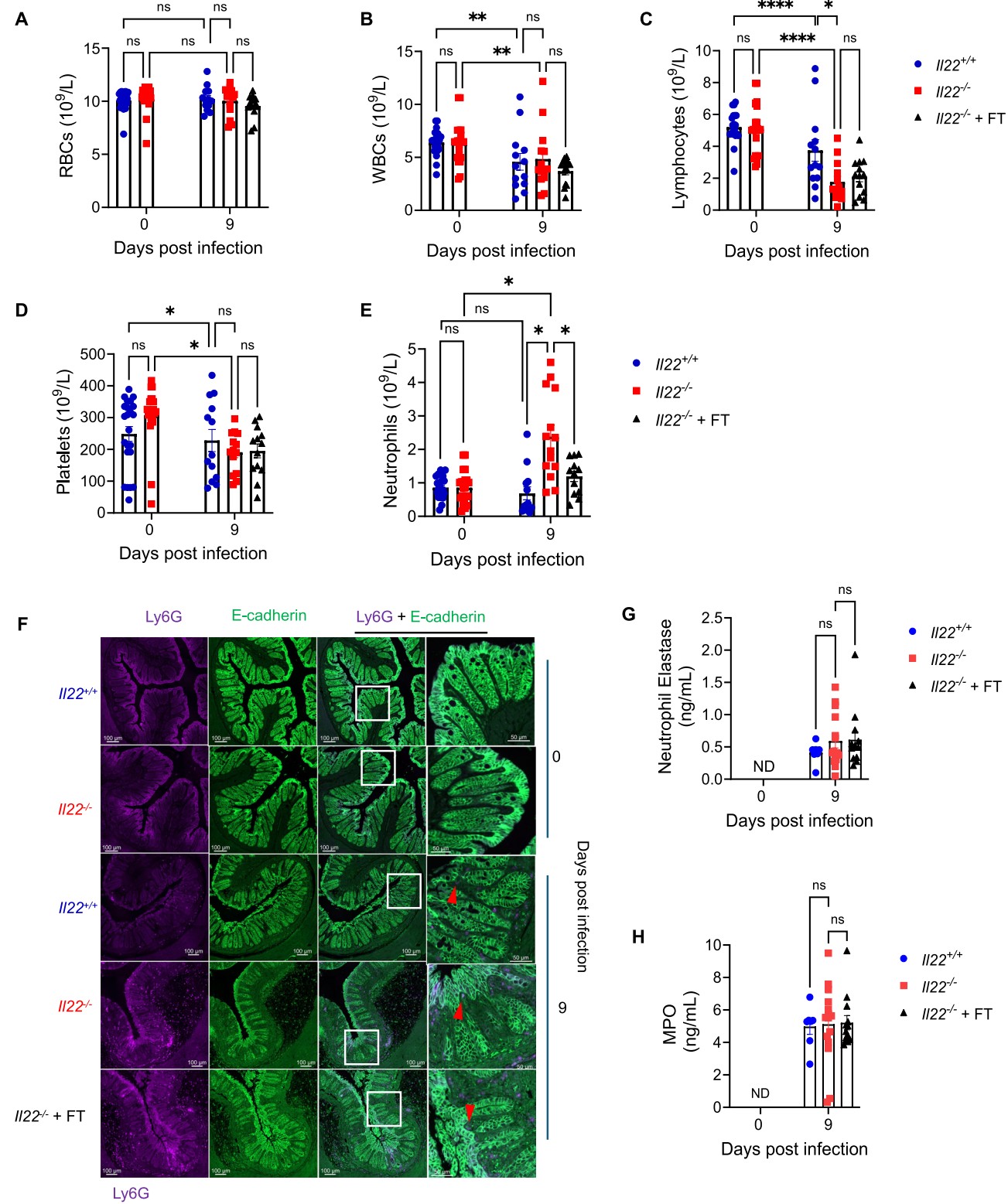

**Fig. 5 | *C. rodentium*-infected *Il22*<sup>−/−</sup> mice display changes in neutrophil dynamics. A–E** Complete blood counts of RBCs (**A**), WBCs (**B**), lymphocytes (**C**), platelets (**D**) and neutrophils (**E**) at 0 and 9 dpi in *Il22*+/+, *Il22*−/− mice ± FT. **F** Representative immunohistochemistry for Ly6G staining neutrophils (violet) and E-cadherin (green) in colonic sections; note the neutrophil staining in the mucosa near the cIECs shown with arrows. Also refer to Fig. S3. Scale bar: 100 μm. **G, H** Neutrophil elastase (**G**) and MPO levels (**H**) in colonic explant cultures. Each dot represents an individual mouse. Data shown are pooled values from 3 biological repeats in (**A–E**) and 2 biological repeats (**G–H**) with 3-5 mice per group. Refer to Supplementary Data 1 for exact group sizes. Data plotted as Mean ± SEM. P values determined by Two-way ANOVA with Bonferroni post-test. ns, non-significant, *, p < 0.05; **, p < 0.01; ****, p < 0.0001. Source data with all raw values and exact p values are provided as a Source Data file.

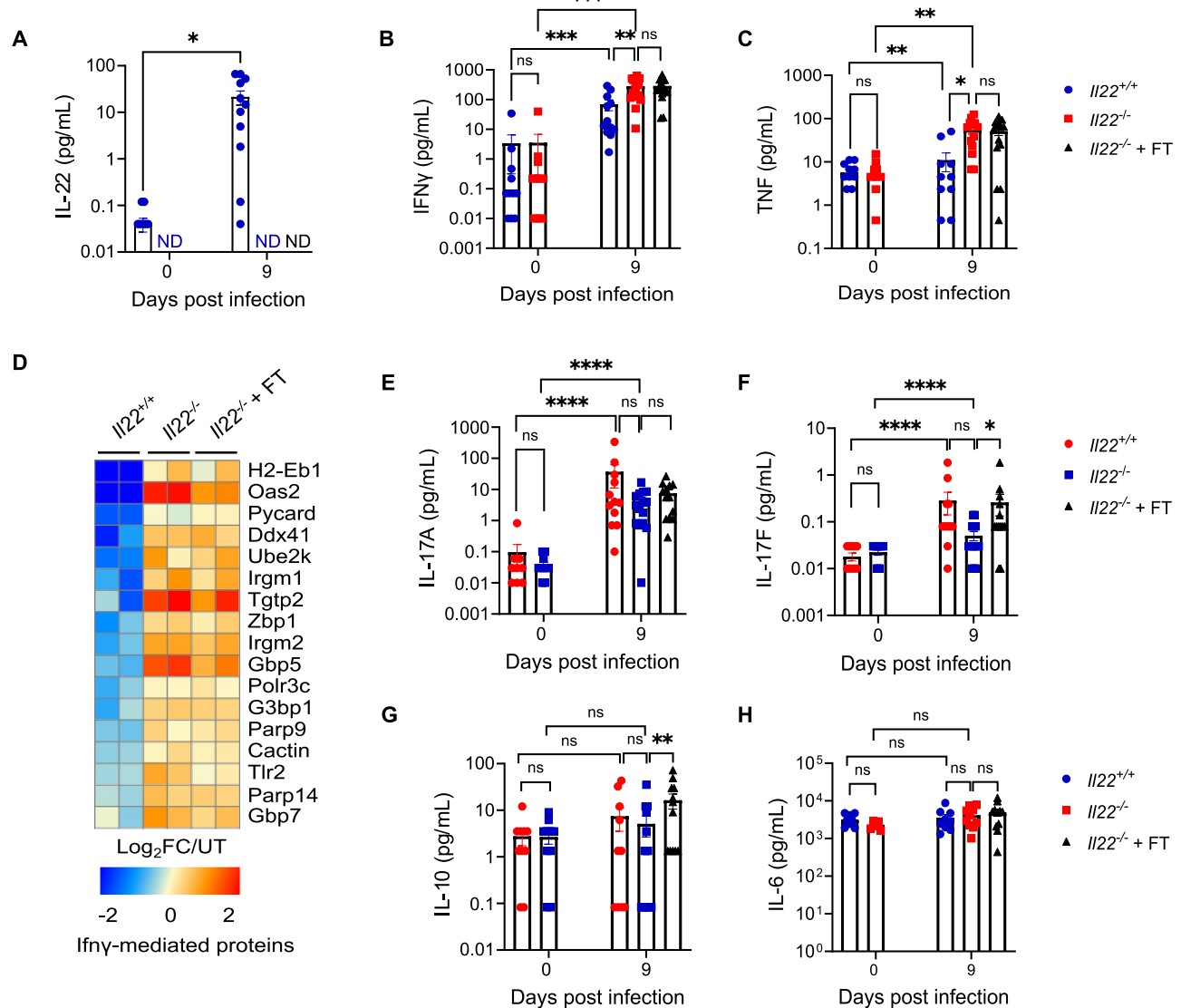

**Fig. 6 | *C. rodentium*-infected *Il22⁻/⁻* mice + FT display higher levels of IL-17F and IL-10. (A-C)** IL-22 **(A)**, IFNγ **(B)** and TNF **(C)** levels in the colonic explant cultures from *Il22⁺/⁺* and *Il22⁻/⁻* mice ± FT at 0 and 9 dpi. **(D)** Heat-map of the proteomic analysis of IFNγ-regulated proteins in colonic IECs from *Il22⁺/⁺* and *Il22⁻/⁻* mice ± FT at 9 dpi. **(E–H)** IL-17A **(E)**, IL-17F **(F)**, IL-10 **(G)** and IL-6 **(H)** levels in the colonic explant cultures at 0 and 9 dpi. Each dot represents an individual mouse. Data shown are pooled values from 3 biological repeats with 3-5 mice per group. Refer to Supplementary Data 1 for exact group sizes. Data plotted as Mean ± SEM. P values determined on log values by Two-way ANOVA with Bonferroni post-test. ns, nonsignificant, *, p < 0.05; **, p < 0.01; ***, p < 0.001; ****, p < 0.0001. Source data with all raw values and exact p values are provided as a Source Data file.

although it remained shorter than in *Il22⁺/⁺* mice (Fig. 7B, C). By 48 dpi, further recovery in colon length was observed.

Histological analysis showed substantial epithelial recovery in FT-treated *Il22⁻/⁻* mice by 20 dpi compared to 9 dpi, although epithelial damage was still greater than in *Il22⁺/⁺* mice (Fig. 7D, E). Both *Il22⁺/⁺* and *Il22⁻/⁻* mice exhibited marked CCH at 20 dpi, but unlike 9 dpi, FT-treated *Il22⁻/⁻* mice had significantly greater CCH than *Il22⁺/⁺* mice (Fig. 7F). By 48 dpi, epithelial architecture and CCH were fully normalised in *C. rodentium*-infected *Il22⁻/⁻* mice, with no discernible differences from *Il22⁺/⁺* controls (Fig. 7D, F).

Colonic IL-22 and IFNγ levels are reported to remain elevated at 48 dpi, about 3 weeks post *C. rodentium* clearance[7,43]. Consistently, *Il22⁺/⁺* mice had higher IL-22 at 20 and 48 dpi, while both *Il22⁺/⁺* and *Il22⁻/⁻* mice showed elevated IFNγ compared to uninfected control mice (Fig. 7G, H). IFNγ levels did not differ between *Il22⁺/⁺* and *Il22⁻/⁻* mice, indicating similar long-term recovery profiles. TNF levels were comparable to uninfected controls, consistent with resolution of inflammation (Fig. S8).

Principal component analysis (PCA) of the quantitative proteomics of colonic IECs showed that unlike at 9 dpi, FT-treated *Il22⁻/⁻* mice clustered with *Il22⁺/⁺* mice at 20 dpi (Fig.7I). Recovered mice also showed restoration of the expression of the ion transporters and their regulators, indicating comparable recovery at the molecular level (Fig. S9). These findings support the conclusion that fluid therapy enables recovery through IL-22-independent mechanisms.

### Recovered *Il22⁻/⁻* mice are protected from *C. rodentium* rechallenge

By 20 dpi, FT-treated *Il22⁻/⁻* mice exhibited substantial colonic epithelial recovery (Fig. 7). To assess systemic recovery, we measured serum parameters and performed CBC analysis. Both *Il22⁺/⁺* and FT-treated *Il22⁻/⁻* mice at 20 dpi had serum protein, corticosterone, renin, and cystatin C levels comparable to uninfected controls (Fig. 8A). Similarly, white blood cell, lymphocyte, platelet, and neutrophil counts did not differ significantly between uninfected and 20 dpi mice, suggesting equivalent systemic recovery in both genotypes (Fig. 8B).

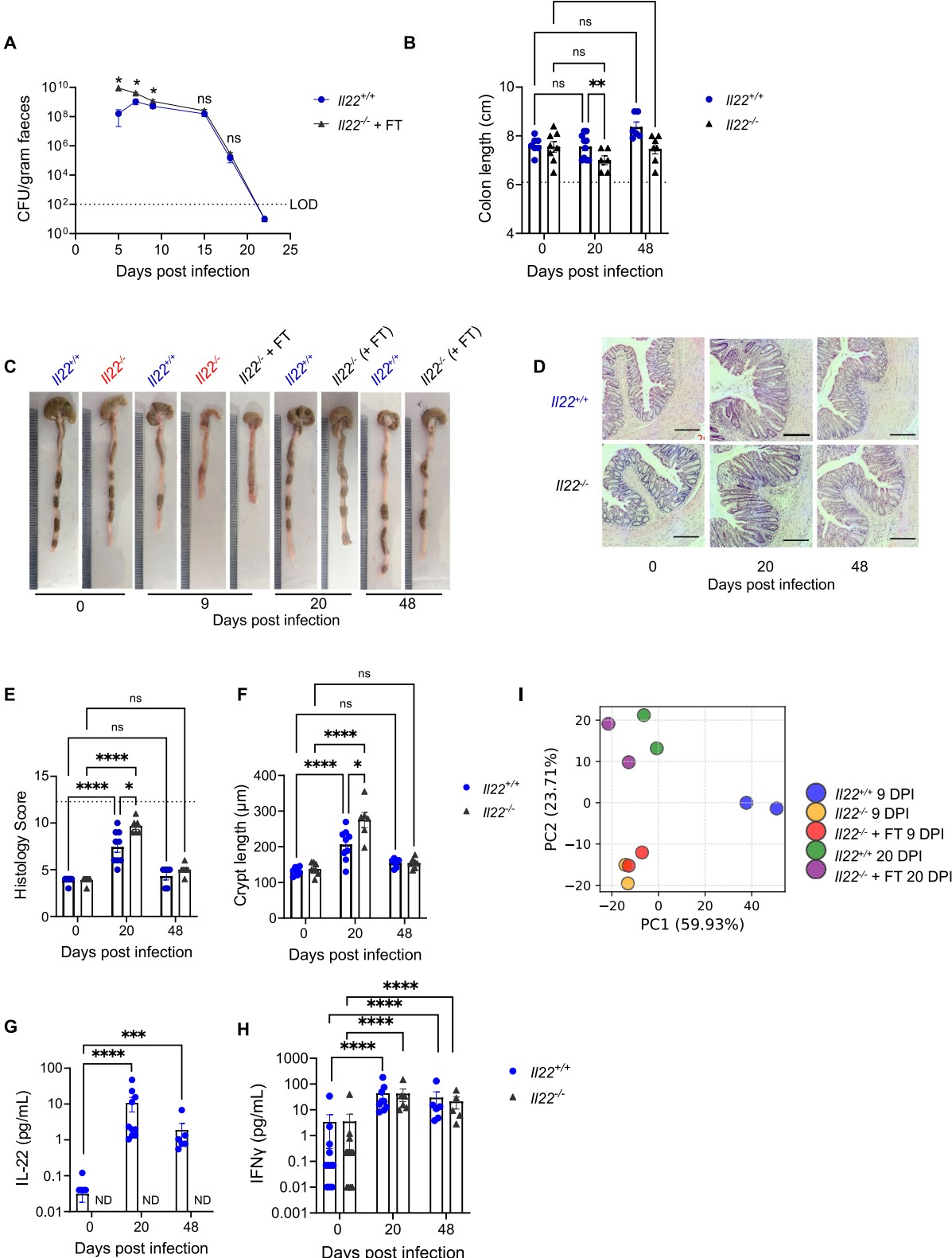

**Fig. 7 | IL-22-independent recovery when dehydration is corrected. A** Faecal CFUs during the peak and clearance phases of *C. rodentium* infection. **B**, **C** Colon length measurements (**B**) and colon images at 20 and 48 dpi (**C**); note the most severe observed colon shortening in *Il22⁻/⁻* mice at 9 dpi and recovery by 20 dpi. **D** Representative H&E-stained colonic sections at 0, 20 and 48 dpi showing recovery. Scale bar: 200 μm. **E**, **F** Histopathology scores (**E**) and CCH measurements (**F**) at 20, and 48 dpi. **G**, **H** IL-22 (**G**) and IFNγ (**H**) levels in the colonic explant cultures at 20 and 48 dpi. (**I**) Principal component analysis of colonic IEC proteomes at 9 and 20 dpi.

Each dot represents an individual mouse. Data shown are pooled values from 2 biological repeats with 3-5 mice per group. Refer to Supplementary Data 1 for exact group sizes. Data for 0 dpi in (**B**, **E**–**H**) is same as used in Figs. 4 and 6, refer to 3Rs section in Methods. Dotted line in (**B**, **E**) represent the mean value for *Il22⁻/⁻* + FT at 9 dpi. Data plotted as Mean ± SEM. P values determined by One-way ANOVA in (**A**) and Two-way ANOVA in (**B**, **E**–**H**) with Bonferroni post-test. ns, non-significant, *, $p < 0.05$; **, $p < 0.01$; ***, $p < 0.001$; ****, $p < 0.0001$. Source data with all raw values and exact p values are provided as a Source Data file.

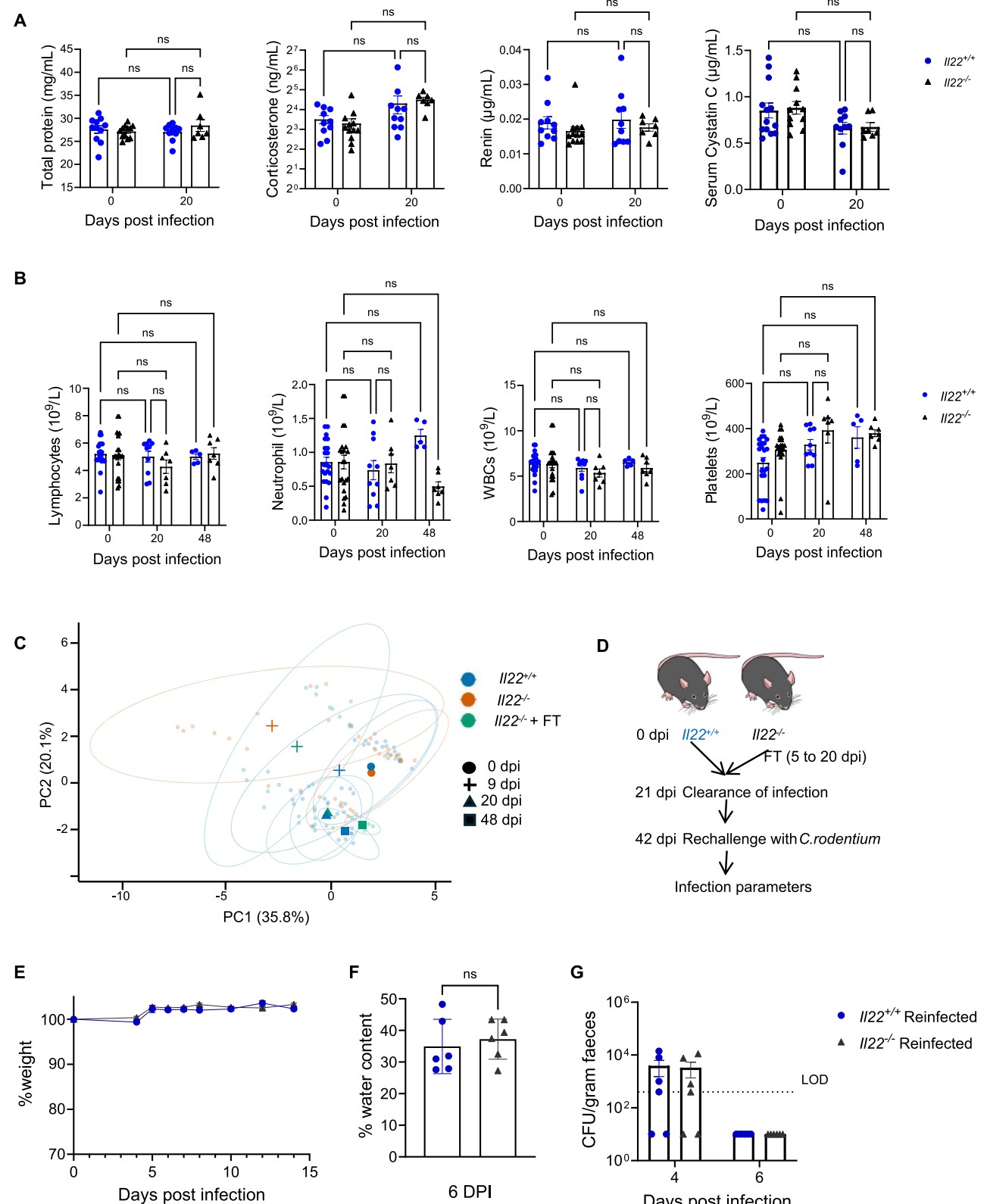

**Fig. 8 | Recovered *Il22*⁻/⁻ mice are protected from *C. rodentium* rechallenge.**
**A** Serum total protein, corticosterone, renin and cystatin C levels at 0 and 20 dpi.
**B** Complete blood counts of lymphocytes, neutrophils, WBCs and platelets at 0, 20
and 48 dpi. **C** PCA analysis of 13 parameters including colonic pathology para-
meters, serum parameters and CBC. **D** Experimental schematic of *C. rodentium*
reinfection after primary infection and recovery with FT. **E** Temporal weight loss
upon *C. rodentium* rechallenge. **F**, **G** Faecal water content **(F)** and CFUs **(G)** during
rechallenge. Each dot represents an individual mouse. Data represent 2 biological
repeats with 3-5 mice per group (see Supplementary Data 1). Data for 0 dpi in **(A, B)**
is the same as used in Figs. 3, 5, refer to the 3Rs section in Methods. Data shown
as Mean ± SEM. P values determined by Two-way ANOVA with Bonferroni post-test
in **(A, B)** and two-tailed Student's t-test in **(F)**. ns, non-significant; *, p < 0.05;
**, p < 0.01; ***, p < 0.001; ****, p < 0.0001. Source data with all raw values and
exact p values are provided as a Source Data file. The mouse cartoon in **(D)** is an
Illustration from NIAID NIH BioArt Source (bioart.niaid.gov/bioart/281).

To holistically assess the disease physiology and recovery, we performed PCA encompassing intestinal pathology, systemic dehydration markers, and peripheral blood counts. While uninfected *Il22*+/+ and *Il22*−/− mice clustered closely, they separated markedly at 9 dpi. FT-treated *Il22*−/− mice clustered closer to *Il22*−/− mice, reflecting the similar pathology observed. Importantly, at 20 and 48 dpi, FT-treated *Il22*−/− mice overlapped with *Il22*+/+ mice indicating a similar recovery (Fig. 8C).

Finally, we tested whether recovered *Il22*−/− mice were protected from *C. rodentium* rechallenge. Mice were reinfected three weeks after clearance of the primary infection (Fig. 8D). Recovered *Il22*−/− mice showed no weight loss upon rechallenge, and faecal water content and faecal CFUs were comparable between *Il22*+/+ and *Il22*−/− mice (Fig. 8E–G), suggesting intact mucosal adaptive immune responses.

## Discussion

Our study demonstrates that mortality in *C. rodentium*-infected *Il22*−/− mice results from dehydration secondary to extensive epithelial damage, rather than from an inability to control bacterial proliferation, systemic dissemination, or epithelial barrier recovery. Although IL-22 is important for maintaining epithelial integrity and limiting infection-induced damage, IL-22-independent pathways can compensate in its absence, enabling bacterial clearance and regeneration of the colonic epithelium when hydration is restored.

Previous studies have established IL-22 as a critical mediator in host defence against *C. rodentium* infection[6,9,10]. *Il22*−/− mice exhibit increased intestinal epithelial damage, higher systemic *C. rodentium* bacterial burden, and 100% mortality. Several studies attributed mortality to IL-22's role in inducing AMPs such as Reg3β and Reg3γ[10,13], essential for bacterial control. However, mice lacking these AMPs individually survived *C. rodentium* infection[28], suggesting that IL-22's protective effects extend beyond AMPs induction. Our findings indicate that the mortality in *Il22*−/− mice is primarily due to diarrhoea, leading to ion loss and subsequent dehydration.

Proteomic and gene expression analyses revealed significantly reduced abundance and expression of key ion transporters and their regulators in *Il22*−/− mice, including *Slc26a3*, *Aqp8*, *Ca2*, *Ca4*, *Slc5a8*, and *Slc15a1*. These proteins are essential for electrolyte and water absorption in the colon[32,44]. The reduced levels of these ion channels do not appear to be directly regulated by IL-22; rather, these transporters are mainly expressed by absorptive enterocytes in the colon, markers of which were expressed at lower levels in *C. rodentium*-infected *Il22*−/− mice. The heightened epithelial injury and inflammation observed in *Il22*−/− mice are likely key drivers of reduced enterocyte populations and subsequent transporter loss. Notably, *Ca4* downregulation has been independently linked to lethality after *C. rodentium* infection and to excessive Wnt signalling in colonic crypts; pharmacological inhibition of Wnt rescued these mice[45]. IL-22 is known to antagonise Wnt signalling[46], which may further contribute to the *Ca4* downregulation observed in *Il22*−/− mice. Furthermore, recent single-cell RNA-sequencing analysis of *C. rodentium*-infected colonocytes identified a distinct subset of absorptive enterocytes directly infected by *C. rodentium*, termed distal colonocytes (DCC)[6]. In this population, ion transporters including *Slc26a3* and *Ca4* were downregulated post-infection, indicating that infected enterocytes sensed *C. rodentium* and altered their gene expression profile. Thus, the further loss of these ion channels in *Il22*−/− mice likely reflects a combined effect of increased enterocyte loss and altered transcriptional responses within infected enterocytes.

The role of hydration status in modulating immune responses has recently gained attention[47,48]. Water-restricted mice, which exhibit lower faecal water content and constipation, display an impaired ability to eliminate *C. rodentium* and reduced immune cell populations in the colon, including B cells, CD4+, and CD8+ T cells[47]. Similarly, downregulation of *Aqp3* is associated with inflammatory and infectious diarrhoea, and *Aqp3*−/− mice exhibit diminished populations of Th17 and Treg cells[47]. Moreover, water- and diet-restricted mice exhibit increased morbidity and mortality following infection with ESKAPE pathogens, including *Klebsiella pneumoniae*, *Acinetobacter baumannii*, *Pseudomonas aeruginosa*, *Enterobacter cloacae*, and *Candida albicans*[49]. In these models, macrophages displayed impaired capacity to phagocytose bacterial pathogens and to efferocytose apoptotic neutrophils.

Implementing a FT regimen effectively mitigated dehydration in *C. rodentium*-infected *Il22*−/− mice, normalising serum dehydration parameters and improving clinical signs without directly affecting bacterial load, systemic dissemination, colonic damage, immune cell population, or expression of major pro-inflammatory markers. FT did not influence epithelial barrier breakdown, faecal bacterial load, or systemic dissemination of *C. rodentium* in *Il22*−/− mice. However, FT-treated *Il22*−/− mice cleared *C. rodentium* infection and survived, demonstrating that once fluid balance is restored, bacterial clearance and tissue recovery can proceed through IL-22-independent pathways, while IL-22 remains critical for preventing fatal dehydration during acute infection in *Il22*+/+ mice.

The rehydration regimen used here was based on veterinary standards for treating dehydration in small rodents[50,51]. Because *Il22*−/− mice display extensive colonic epithelial damage and loss of absorptive transporters, oral rehydration would be physiologically ineffective and technically impractical at the required volumes. Subcutaneous administration of a balanced electrolyte solution therefore represents the most appropriate method for restoring systemic hydration in this model.

Although most studies of *C. rodentium*-mediated pathology have focused on the colon[3], *C. rodentium* infection also induced leukopenia, lymphopenia, and thrombocytopenia in both *Il22*+/+ and *Il22*−/− mice, indicating systemic effects with a substantial impact on the haematopoietic system. Consistently, it has been previously shown that bone marrow neutrophils from *Il22*+/+ C57BL/6 mice show significant changes and upregulation of activation markers upon *C. rodentium* infection[52]. Furthermore, recent reports suggest that *C. rodentium* infection in *Il22*+/+ mice affects bone remodelling, leading to compromised bone architecture due to elevated circulating TNF levels post-infection[53]. Despite changes in total blood cell counts, *Il22*+/+ mice exhibited no differences in circulating neutrophil levels, consistent with previous reports[52]. In contrast, *Il22*−/− mice displayed peripheral neutrophilia, indicative of a heightened inflammatory state. Similarly, while overall immune cell populations in the colon were comparable between *Il22*−/− and *Il22*+/+ mice, *Il22*−/− mice harboured higher colonic neutrophil numbers, suggesting increased local inflammation and tissue damage, as reported recently[28].

IL-17A, IL-17F, IL-6, and IL-10 perform overlapping functions when IL-22 is limited[41,42,54,55]. *Il22*−/− mice, with or without FT, displayed induction of IL-17A, IL-17F, and IL-10 upon infection. However, FT-treated *Il22*−/− mice exhibited significantly higher IL-17F and IL-10 levels compared to untreated controls, suggesting a plausible mechanism for IL-22-independent recovery. IL-17A and IL-17F, co-produced with IL-22 by Th17 cells, enhance the production of AMPs such as Reg3β and Reg3γ[41,42,56], reinforcing epithelial defence mechanisms. FT-treated mice displayed higher *Reg3g* expression, indicating a possible functional overlapping role of the higher IL-17F in the absence of IL-22. IL-6 signals through the gp130 receptor complex to activate STAT3 in intestinal epithelial cells[54], promoting cell survival, proliferation, and tissue repair. Additionally, IL-10, known for its potent anti-inflammatory properties[55,57], tempers excessive inflammation and further supports mucosal homeostasis. Collectively, these cytokines form a network that may lead to epithelial recovery in the absence of IL-22.

With adequate treatment of dehydration, *Il22*−/− mice exhibited substantial recovery of colonic architecture and function by 20 dpi, achieving complete restoration by 48 dpi. Recovered mice also

showed normalisation of serum parameters and circulating blood cell counts, indicating resolution of systemic disease and inflammation. Interestingly, post-recovery *Il22*[-/-] mice displayed features of prolonged convalescence similar to those observed in *Il22*[+/+] mice, including persistently elevated colonic IFNγ levels. Furthermore, recovered *Il22*[-/-] mice were protected from rechallenge with *C. rodentium*, indicating that mucosal adaptive immunity remains intact in the absence of IL-22.

The quantitative proteomics data in this study were generated from pooled IECs from two independent experiments and were therefore exploratory, providing molecular context for the observed physiological and immunological phenotypes rather than forming the basis for any conclusion or deciphering mechanisms. Future studies with higher replicate numbers will enable deeper statistical resolution and assessment for IL-22 independent recovery mechanisms.

Taken together, our data demonstrate that IL-22 safeguards survival during enteric infection by preventing fatal dehydration. When hydration status is maintained, IL-22-independent pathways are sufficient to mediate bacterial clearance and intestinal recovery. While FT did not restore epithelial ion transporters or barrier integrity during the acute phase, it permitted survival and subsequent repair, indicating activation of an IL-22-independent mechanism. Future studies will aim to dissect the IL-22-independent pathways, including the potential roles of IL-17F, IL-10.

In conclusion, the present findings establish dehydration as the proximate cause of mortality in *Il22*[-/-] mice, representing a paradigm shift in understanding IL-22−dependent host protection. The administration of supportive therapy is common in clinical care but is rarely applied to animal models of disease. Our findings also highlight that integrating such measures enhances the physiological realism of murine models and their relevance to model human disease.

## Methods

### Mouse experiments

Mouse experiments were performed in accordance with the Animals Scientific Procedures Act of 1986 and were approved by the local Ethical Review Committee according to UK Home Office guidelines. The *Il22*[+/+] and *Il22*[-/-] mice from same founder mice were maintained in homozygous condition and were housed and bred in dedicated animal facilities of Imperial College London (12 h light/dark cycle; 22 ± 2 °C; 30 to 40% humidity). Mice were housed in IVC cages with corn cob bedding and enrichments including nesting material, refuges, and gnawing sticks. Mice were fed with RM1(E) rodent diet (SDS Diet) and water ad libitum. Mice (age 2-4 months, 25 to 30 grams weight) were used for experiments. All mice were genotyped and tested for the presence or absence of iCre using multiplex PCR as described[58]. The sequences of primers (5' to 3') used: Forward, CAGGCTCTCCTCTCAGTTATCA; Wildtype reverse, TCCTGAAGGCCAAAATAGG; Mutant reverse, CCTCAGGTTCAGCAGGG AAC.

### *C. rodentium* infection

*C. rodentium* (strain ICC169) was grown overnight in lysogeny broth at 37 °C with shaking at 180 rpm, centrifuged at 2500 x *g* for 10 min and resuspended in sterile PBS. Mice were infected with approximately 1 ×109 CFU in 200 μL sterile PBS using oral gavage, as previously described[1]. Mock infected (uninfected/0 dpi) mice received 200 μL sterile PBS. The inoculum CFU was confirmed by CFU quantification as previously described[1]. Mice were monitored every day for changes in weight and disease severity was scored for four parameters: coat fur, posture, skin turgor and mobility; scores ranged from 0 to 5, wherein 0 scores for a healthy mouse. Humane end points were met if a mouse lost 20% of its body weight or if a disease score of 3 or more was observed for more than 2 consecutive days.

For rIL-22 treatment of naïve mice, mice were injected with 2 μg of rIL-22 (Peprotech; catalogue number 210-22), intraperitoneally for two consecutive days. On day 3, mice were ethically euthanised and the distal colon was harvested for total RNA preparations.

### Measurement of faecal trait score faecal water, Sodium and Potassium content and *C. rodentium* shedding

Faecal traits were assessed using the Bristol stool scoring system adapted for laboratory animals[31]. Briefly, stool consistency is graded as follows: type 1, hard pellets resembling nuts; type 2, firm and lumpy sausage-shaped stool; type 3, sausage-shaped with surface cracks; type 4, smooth, soft, and snake-like; type 5, soft clumps with well-defined edges; type 6, mushy stool with ragged edges; and type 7, entirely liquid with no solid components.

To determine the faecal water content, faeces were freshly collected in pre-weighed 1.5 mL tubes with punctured cap. The tubes with wet faeces were weighed immediately and incubated at 55 °C. The tubes were weighed everyday till the weight did not change and were recorded. The wet weight and dry weight of faeces was determined by subtracting the weight of the tube and faecal water content was estimated using the following equation:

$$\%water\ content = \frac{wet\ weight - dry\ weight}{wet\ weight} x100$$

Faecal sodium and potassium were estimated as described earlier[59]. Briefly, dry faeces were resuspended in autoclaved double distilled water, homogenised and centrifuged at 100 x *g* for 1 min. The sodium and potassium content in the supernatant was estimated using LAQUAtwin metres (Horiba, Japan) and was normalised to the dry weight of faeces.

To determine *C. rodentium* faecal shedding, fresh faecal samples were collected at the specified dpi, homogenised in sterile PBS, serially diluted and plated on LB-agar containing 50 mg/kg of kanamycin. For rechallenge experiments, mice were infected with approximately 1 ×109 CFU of *C. rodentium*, orally at 50 dpi post primary infection.

### Post-mortem pathophysiological analysis

Mice were anaesthetized via intraperitoneal injection of ketamine (100 mg/kg) and medetomidine (1 mg/kg). Once pedal reflexes were absent, they were positioned in dorsal recumbency. Blood was drawn from the right ventricle using a closed approach, inserting a 25 G needle at a 30° angle to the skin until free aspiration into a 1 ml syringe was achieved. Immediately after blood collection, mice were euthanised by cervical dislocation. The blood was collected in EDTA coated tubes for CBC analysis and in serum separator tubes (SST) for serum parameter analysis. The blood in SST was left to clot at RT for 1 h before centrifugation at 20,000 × *g* for 3 min. Serum was then aliquoted and stored at -80 °C until further use.

Euthanised mice were dissected and the large intestine of mouse consisting of caecum and colon was harvested and laid down on a clean surface in parallel to a mm scale to estimate the colon length and was captured using a digital camera. Colon was removed from caecum, cleaned and weighed using a digital scale. The weight of the colon was normalised for its length and recorded. From the distal side of colon, 0.5 cm colon was stored in 4% paraformaldehyde for histological studies, next 0.5 cm was stored in RNAlater for total RNA preparation and 3.5 cm was collected for colonic IECs preparation. The liver and spleen were harvested, weighed and homogenised in sterile PBS for estimation of systemic burden of *C. rodentium*.

### CBC and Serum parameters

CBC analysis of the anticoagulated blood was performed using the HM5 Vetscan analyser according to the manufacturer's instruction.

Total protein concentration in serum was estimated using a NanoDrop spectrophotometer (Thermo Scientific). 2 μl of serum sample was used to measure absorbance at 280 nm and the corresponding

protein concentration was recorded. Serum renin, corticosterone and Cystatin C concentration were determined using Mouse Renin 1 ELISA Kit (ThermoFischer Scientific; catalogue number EMREN1), Corticosterone Parameter Assay Kit (Bio-techne; catalogue number KGE009) and Mouse Cystatin C DuoSet ELISA (Bio-techne; catalogue number DY1238), respectively according to manufacturer's protocol.

### FT protocol

Infected *Il22*$^{-/-}$ mice were administered with a FT regimen 5 to 20 dpi, where they were provided with wet food and three subcutaneous injections of balanced electrolyte solution, daily. To prepare wet food, RM1(E) (SDS Diet) dry food pellets (average weight of 1.1 gram/pellet) were soaked in drinking water in 50 mL tubes for 30 min, excess fluid was drained and one pellet per mouse was provided in sterile 6 cm dishes inside the cage. For subcutaneous injections, balance electrolyte solution[60] (PlasmaLyte 148, Baxter International Inc.) was pre-warmed to 37 °C and the dose was determined based on dehydration status of mice. The dehydration status of mouse was calculated based on scoring the clinical parameters of Anorexia- a lack of appetite and body weight loss, reduced motility, coat and posture, scores ranged from 0 to 5, wherein 0 scores for a healthy mouse. The dose of injections was calculated using following equation:

$$Body\ weight\ (grams) \times \%\ Dehydration\ (as\ a\ decimal\ value) = Fluid\ volume\ (ml)$$

The volume of fluid to be administered was divided into 3 injections per day. Refer to Table S3 for example.

### Histological analysis and immunofluorescence staining

Distal colon samples (0.5 cm) were fixed in paraformaldehyde, paraffin-embedded, and sectioned at 5 µm thickness. Sections were stained with haematoxylin and eosin (H&E) or processed for immunofluorescence. H&E-stained slides were imaged using a Zeiss AxioVision Z1 microscope equipped with a 20X objective and an AxioCam MRm camera, and images were processed with Zen 2.3 software (Blue edition; Carl Zeiss MicroImaging GmbH, Germany). Histopathological scoring was performed following a protocol adapted from Bleich et al., 2004[61]. Three parameters were evaluated in each section: (1) extent of epithelial damage and inflammation, (2) degree of hyperplasia, and (3) percentage of area affected. Each was scored on a scale of 0–5, with 0 representing normal histology. CCH measurements were carried out on H&E-stained sections as described previously[1,7].

For immunofluorescence staining, tissue sections were dewaxed by sequential submersion in Histo-Clear (2×10 min), 100% ethanol (2×10 min), 95% ethanol (2×10 min), 80% ethanol (2×3 min), and PBS-TS (PBS + 0.1% Tween 20 + 0.1% saponin, 2×3 min). Antigen retrieval was performed by heating sections for 30 min in demasking solution (0.3% trisodium citrate + 0.05% Tween 20, pH 6.0), followed by cooling. Slides were blocked in PBS-TS with 10% normal donkey serum for 1 h in a humid chamber, then incubated overnight at 4 °C with respective primary antibody (anti-E-cadherin, Abcam; catalogue number ab76055, 1:100; anti-PCNA, Abcam; catalogue number ab29, 1:500; polyclonal anti-*C. rodentium*, Statens Serum Institute, Copenhagen, Denmark, 1:100; anti-Ly6G, Biolegend; catalogue number 127602, 1:100) in PBS-TS with 10% donkey serum. After rinsing twice (10 min each) in PBS-TS, sections were incubated with Alexa Fluor 488 labelled anti-mouse or Alexa Fluor 647 anti-rabbit IgG secondary antibody (Jackson ImmunoResearch; 715-545-150, 1:100) and DAPI (1:1000), followed by additional washes. Slides were mounted using ProLong Gold antifade and cured overnight in the dark. All immunofluorescence images were acquired using a Zeiss AxioVision Z1 microscope with a 20X or 40X lens objective using a Hamamatsu C11440 digital camera and processed using Zen 2.3 (Blue version; Carl Zeiss MicroImaging GmbH, Germany). The number of neutrophils in

mucosa and sub-mucosa were calculated as described earlier[52]. Briefly, the number of Ly6G$^+$ E-cadherin$^-$ DAPI$^+$ cells per mm$^2$ were quantified by counting the cells in each region and diving the number by the total area of the region, calculated using the Zen 2.3 software.

### FITC−dextran intestinal permeability assay

Intestinal permeability was assessed using fluorescein isothiocyanate (FITC)-dextran (4 kDa; Sigma-Aldrich; catalogue number FD4). FITC-dextran was administered by oral gavage at a dose of 440 mg/kg body weight (dissolved in sterile PBS). After 4 h, blood was collected in SST by cardiac puncture, and serum was separated as described above. Serum FITC-dextran concentrations were measured in duplicate using a fluorescence plate reader (excitation 485 nm, emission 528 nm) against a standard curve.

### Cytokine profiling explants

0.5-cm distal colonic tissue sample without faeces was weighed and incubated for 2 h in RPMI 1640 medium with glutamine (Sigma) with 100 µg/mL of streptomycin and 100 µg/mL of penicillin. Following this, explants were cultured in complete RPMI medium at 1 mL/0.1 g tissue (RPMI 1640 medium with glutamine supplemented with 10 % heat-inactivated FBS, 1 mM sodium pyruvate, 100 µg/mL penicillin, 100 µg/mL streptomycin, and 10 mM HEPES) for 24 h at 37 °C with 5% $CO_2$. The supernatant was extracted, centrifuged for 10 min at 3,000 x g to remove cellular debris, and stored at −80 °C until further analysis.

Explant cytokine levels were assessed from the corresponding samples via LEGENDplex kit (BioLegend, catalogue number 741044) as per manufacturer's instructions. Cytokine levels were acquired using a FACSCalibur flow cytometer (BD Biosciences), and analyses were performed using LEGENDplex data analysis software (BioLegend). Explant NE and MPO were assessed using the NE ELISA kit (Bio-techne; catalogue number DY4517) and MPO ELISA kit (Bio-techne; catalogue number DY3667), according to manufacturer's protocol.

### Isolation of colonic immune cells and flow cytometry analysis

Colonic immune cells were isolated from mouse colons as described[62]. Briefly, 3-cm segments of distal colon were excised, washed, and cut opened longitudinally, incubated at 37 °C for 20 min in a shaking incubator in calcium- and magnesium-free 1X HBSS containing 2% FBS, 10 mM EDTA, and 1 mM DTT. Following incubation, the cell suspension was centrifuged to separate IECs. Supernatants containing IECs were discarded, and the residual tissue was subjected to enzymatic digestion in RPMI 1640 medium containing 62.5 µg/mL Liberase, 50 µg/mL DNase I (Sigma-Aldrich), and 2% FBS at 37 °C for 40−50 min, followed by filtration through 100 µm cell strainer to obtain a single-cell suspension for flow cytometry analysis.

For extracellular staining, single-cell suspensions were incubated for 10 min with LIVE/DEAD Fixable Blue dye to identify and exclude non-viable cells from analysis. Following this, cells were incubated for 20 min with Fcγ receptor blocking reagent (BD Biosciences) to prevent non-specific antibody binding, then stained for surface antigens using fluorochrome-conjugated monoclonal antibodies (Refer to reagents table). All staining procedures were carried out at 4 °C in the dark. Unstained controls, LIVE/DEAD-only controls, and fluorescent-minus-one (FMO) controls were included to assess background fluorescence and set gating thresholds. Post staining, cells were washed and fixed for 20 min using the eBioscience Foxp3/transcription factor fixation buffer set. Fixed samples were stored at 4 °C until acquisition.

For compensation, single-colour controls were prepared using UltraComp eBeads™ Compensation Beads. Prior to analysis, cells were washed and resuspended, and data were acquired on 50,000 live events per sample using an Aurora flow cytometer (Cytek Biosciences). Flow cytometric data were analysed with FlowJo software (version 10.8.1, Tree Star). The gating strategy used to identify various cell subsets is illustrated in Fig. S5A.

## Colonic IECs purification for proteomics analysis

Colonic IECs were isolated from 3.5-cm distal colonic tissue samples, as previously described[1]. Briefly, the 3.5-cm colonic tissue sample was opened longitudinally and washed in 1X Hanks' balanced salt solution (HBSS) without Magnesium ($Mg^{+2}$) and Calcium ($Ca^{+2}$). The tissue sample was incubated at 37 °C with shaking for 45 min in enterocyte dissociation buffer (1X HBSS without $Mg^{+2}$ and $Ca^{+2}$, containing 10 mM HEPES, 1 mM EDTA, and 5 μL/mL 2-β-mercaptoethanol). The remaining tissue was removed, and lifted enterocytes were subsequently collected by centrifugation (2000 x $g$ for 10 min), followed by three PBS washes at 4 °C. Colonic IEC pellets were stored at -80 °C, till further use.

## Sample preparation and TMT labelling

*C. rodentium*-infected *Il22*$^{+/+}$ or *Il22*$^{-/-}$ mice and corresponding uninfected controls were lysed using probe sonication in TEAB (triethylammonium bicarbonate) buffer (100 mM TEAB, 1% SDC, 1% isopropanol, 50 mM NaCl) containing protease and phosphatase inhibitors, then sonicated again before protein quantification using the Bradford assay (BioRad). Three to four mice per replicate were pooled in 1:1 ratio per sample, reduced and alkylated with 5 mM tris(2-carboxyethyl)phosphine (TCEP) and 10 mM iodoacetamide (IAA) for 30 min. Total of 60 ug of each of protein lysate from each condition were digested with trypsin (75 ng/μL, 18 hrs, RT) and labelled with TMTpro 18-plex reagents (Thermo Scientific) according to manufacturer instructions. SDC was precipitated with 2% formic acid and removed by centrifugation (10,000 rpm, 5 min). TMT-labelled peptides were dried using a vacuum concentrator.

## High-pH RP fractionation and LC-MS analysis

TMT-labelled peptides were fractionated by high-pH RP-HPLC using a C18 column (Waters XBridge) with a gradient elution (5–80% mobile phase B; 0.1% ammonium hydroxide in acetonitrile) at 200 μL/min. Ninety-two fractions were concatenated into 24, dried, and reconstituted in 0.1% formic acid. Approximately 3 μg of peptides per fraction were analysed on an Orbitrap Ascend mass spectrometer using a Real-Time Search-SPS-MS3 method. Peptides were separated on a nanocapillary column (Acclaim PepMap C18, 75 μm × 50 cm, 2 μm, 100 Å, 120-min gradient) at 50 °C. MS1 scans (400–1600 m/z) were acquired at 120,000 resolutions with a dynamic exclusion of 45 s, with standard AGC and auto injection time, and included 2-6 precursor charge states. MS2 scans were acquired in the ion trap using HCD (32% CE). SPS-MS3 scans (HCD CE 55%) were collected at 45,000 resolutions for quantification. These spectra were searched against the *Mus musculus* and *C. rodentium* reviewed proteomes using the Comet search engine, with parameters set for tryptic peptides allowing a maximum of one missed cleavage. Static modifications included cysteine carbamidomethylation (+57.0215) and N-terminal/lysine TMTpro (+304.2071), with variable modifications including Asn/Gln deamidation (+0.984) and Met oxidation (+15.9949), allowing up to two variable modifications and a maximum of four peptides per protein. Precursors meeting these criteria were selected for SPS10-MS3 scans, performed at an Orbitrap resolution of 45,000 with normalised HCD collision energy set to 55%, AGC at 200%, and a maximum injection time of 200 ms.

The mass spectrometry proteomics data have been deposited to the ProteomeXchange Consortium via the PRIDE[63] partner repository with the dataset identifier PXD061225.

## Proteomics data analysis

Proteome Discoverer 3.0 (Thermo Scientific) was used with SequestHT and Comet search engines for protein identification and quantification. Spectra were searched against *Mus musculus* and *C. rodentium* protein entries in UniProt, with a precursor mass tolerance of 20 ppm and a fragment mass tolerance of 0.5 Da. Fully tryptic peptides were considered, allowing up to 2 missed cleavages. Static modifications were TMT at N-termini/K and carbamidomethylation at C residues.

Dynamic modifications included oxidation of methionine, and deamidation of N/Q.

Peptide confidence was estimated using Percolator, maintaining a false discovery rate (FDR) of 1% with target-decoy database validation. Quantification was performed using the TMT quantifier node, with a 15 ppm integration window and the most confident centroid peak at MS2 level. Only unique peptides were used for quantification, with a signal-to-noise ratio threshold of >3. Proteomic data were median normalised to account for variations in sample loading. Log2 fold changes (log2FC) were calculated by comparing *C. rodentium*-infected samples to their corresponding uninfected controls. To correct for potential batch effects, the data were scaled within each biological group. Missing values in the $log_2$-transformed TMT intensity matrix were imputed using a Gaussian distribution-based approach implemented in Perseus. Specifically, values were drawn from a normal distribution with a mean shifted 1.8 standard deviations below the global mean intensity and a width of 0.3 standard deviations. This approach assumes that missing values predominantly represent low-abundance peptides falling below the limit of detection and helps minimise artefactual variance in downstream statistical analyses. Differential protein expression was evaluated using two-sample t-tests in Perseus (version 2.0.11). Due to the limited number of biological replicates (n = 2), FDR correction was not applied, as this eliminated all hits. Proteins were considered differentially expressed if they met a nominal significance threshold ($p < 0.05$) and an effect size filter of |$log_2$ fold change | ≥ 0.7, corresponding to a change approximately threefold greater than the median coefficient of variation. Enrichment analysis of significantly regulated proteins was conducted using Fisher's exact test, applying Benjamini-Hochberg correction to control the false discovery rate (FDR < 0.05). Visualisations of significantly enrichment terms were generated using the Matplotlib library in Python.

## Quantitative reverse transcription-PCR (qRT-PCR)

0.5 cm of distal colon was harvested and incubated in RNAlater for 24 h at 4 °C. Total RNA extraction was performed using the RNeasy kit (Qiagen) according to the manufacturer's protocol. Reverse transcription used 2 μg of purified RNA and High-capacity cDNA Reverse Transcription kit. Quantitative PCR was performed using SsoAdvanced Universal SYBR green supermix on a StepOnePlus Real-Time PCR system. Reactions were performed in duplicate, including negative control lacking cDNA. The ΔCT method of quantification was performed to give values relative to the expression of the housekeeping gene *Gapdh*. Primer pairs used are listed in Table S4.

## Analysis of publicly available gene expression data

Cell type–specific expression of the selected ion channels was assessed using single-cell RNA sequencing data from Avraham-Davidi et al. 2025[34], available on the Single Cell Portal[64] (The Broad Institute of MIT and Harvard; accession SCP1891). Gene expression patterns were visualised using the portal's dot plot analysis tool.

To evaluate the effect of rIL-22 treatment on ion channel expression, we analysed gene expression data from mouse colonic organoids reported by Powell et al., 2020[33]. Extracted expression values, along with associated p-values, are provided in Table S2.

## Statistical analysis and data plotting

No statistical methods were used to determine sample size. Experiments were planned as randomised blocks with 3-5 animals per genotype per experiment, and experiments were repeated independently at least 2 times. The details of number of mice used for all experiments are mentioned in Supplementary Data 1. Data from all mice in all experiments were pooled and analysed. Data were plotted as Mean ± SEM.

Statistical significance between two normally distributed groups was determined by Student's *t*-test. ANOVA was used to test statistical significance for more than two normally distributed groups. When data

were not normally distributed (based on D'Agostino-Pearson or Shapiro-Wilk normality tests), a logarithm transformation was applied, and they were then analysed by ANOVA. Statistical analysis of proteomics data is detailed in the section above. False discovery rate (FDR; Q = 5 %) was used to correct for multiple comparisons (Bonferroni correction) as implemented in GraphPad Prism 10.4.0. FDR-adjusted p values > 0.05 were considered non-significant (ns). Data plotting and statistical analysis were performed using Prism 10.4.0 (GraphPad Software Inc.). Statistical details of experiments are described in the figure legends.

### Ethical statement and implementation of the 3Rs

All animal work was conducted at Imperial College London (Association for Assessment and Accreditation of Laboratory Animal Care accredited unit) under the auspices of the Animals (Scientific Procedures) Act (UK) 1986[65] (PP7392693). The animal experiments were approved locally and were designed in agreement with the ARRIVE[66] guidelines and Imperial College London's animal welfare policies, which are founded on the principles of Replacement, Reduction, and Refinement (3Rs). Accordingly, to minimise animal use, baseline (0 dpi) data from $Il22^{+/+}$ and $Il22^{-/-}$ mice were used as shared controls for multiple analyses, as indicated in the figure legends. This approach reduced unnecessary duplication of control groups while maintaining statistical power and experimental rigour.

### Significance statement

IL-22 is considered essential for host survival during *Citrobacter rodentium* infection, yet the cause of mortality was inconclusive. Here, we show that $Il22^{-/-}$ mice succumb due to dehydration, and that fluid therapy prevents mortality, revealing IL-22-independent recovery mechanisms. Our findings reveal that IL-22 indirectly maintains hydration during *C. rodentium* infection by limiting infection-induced damage, thereby preventing diarrhoea and dehydration.

### Ethical statement

All animal work was conducted at Imperial College London (Association for Assessment and Accreditation of Laboratory Animal Care accredited unit) under the auspices of the Animals (Scientific Procedures) Act (UK) 1986 (PP7392693). Work was approved locally by the institutional ethics committee (AWERB).

### Reporting summary

Further information on research design is available in the Nature Portfolio Reporting Summary linked to this article.

## Data availability

All the materials and key resources can be found in the "Reagent and Resources table" on Figshare: https://doi.org/10.6084/m9.figshare.29980600.v2. The mass spectrometry proteomics data have been deposited in the ProteomeXchange Consortium via the PRIDE[63] partner repository with the dataset identifier PXD061225. Source data are provided with this paper. The source data with all raw values for graphs and plot used in this study and raw abundances of proteins from the proteomics analysis are available at https://doi.org/10.6084/m9.figshare.29980600.v2. Source data are provided with this paper.

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

## Acknowledgements

We thank Prof. Brigitta Stockinger, The Francis Crick Institute, for providing the *Il22⁻/⁻* mice used to establish the colony at Imperial College. We thank Dr. Julia Sanchez-Garrido for careful reading of the paper and critical suggestions, and the staff of Central Biomedical Services, Imperial, for assistance in maintenance and breeding of animals. This work was supported by Wellcome Trust Investigator Award grants 107057/z/15/z and 224282/Z/21/Z.

## Author contributions

Conceptualization: V.M. and G.F. Methodology: V.M., P.B., J.L.C.W., Z.K. Formal analysis: V.M., P.B., Z.K. Investigation: V.M., P.B., Z.K., J.C. Writing–original and draft: V.M., Z.K., and G.F. Writing review and editing: V.M., P.B., Z.K., J.L.C.W., J.C., and G.F. Visualisation: V.M., P.B., Z.K. Supervision: V.M., J.C., and G.F. Funding *acquisition*: G.F.

## Competing interests

The authors declare no competing interests.
