## [Transparent Peer Review file · Nature Communications]

Rehydration rescues Il22^{-/-} mice from lethal *Citrobacter rodentium* infection

Corresponding Author: Dr Gad Frankel

Version 0:

Reviewer comments:

Reviewer #1

(Remarks to the Author)

In this manuscript Mishra and colleagues explore the potential role of dehydration in mortality in *C. rodentium* (Cr)-infected Il22^{-/-} mice. The authors report that the administration of fluids for 2 weeks post-infection completely rescues Il22^{-/-} mice. They suggest that the deaths of these mice are due to dehydration rather than bacterial burden and tissue recovery failure. However, there are several concerns that require addressing: 1) Discrepancy in disease severity compared to historical data, 2) Inconsistencies in some figures, and 3) Absence of quantitative histological analyses and controls in key experiments.

Major Concerns:

Major Point 1: Disease Severity Discrepancy

The severity of phenotypes in both WT and Il22^{-/-} mice after infection appears less pronounced than historical data suggests. Il22^{-/-} mice begin to succumb to infection by day 7 in several reported studies using the Cr model (Zheng et al., 2008, Nature Medicine; Zindl et al., 2022, Immunity), whereas in this study Il22^{-/-} mice do not succumb to infection until day 10. This difference may be attributed to variations in the *C. rodentium* strain used, experimental protocol, or differences in gut microbiota composition. Additionally, within this manuscript there are major differences in severity as shown when comparing Il22^{-/-} mice survival data in Fig. 1B to Il22^{-/-} mice survival data in Fig. 3G. In Fig. 1B, mice survive until day ~14 whereas all mice succumb by day 10 in Fig. 3G. Furthermore, the mild diarrhea in WT mice and the delayed mortality in Il22^{-/-} mice could indicate an underlying SFB infection or another pathobiont, potentially leading to milder symptoms and delayed peak of bacterial burden (Fig. 1B-D). Fig. 1C shows no diarrhea in Il22^{+/+} mice when it is widely reported that wild-type mice also develop clinical symptoms of acute Cr infection. The attenuated phenotype seen in these mice is a potential confounding variable in assessing the relevance of fluid resuscitation for rescuing Il22^{-/-} mice and the claim by the authors that IL-22 is dispensable in the Cr model. In addition, the colon pathology, including hyperplasia, appears similar between infected WT and Il22^{-/-} mice (Figs. 4C and 5D). It is conceivable that fluid resuscitation would improve the hemodynamic parameters of Il22^{-/-} mice that have overwhelming bacterial burden, as has been previously reported to occur in Il22^{-/-} mice. The authors here suggest improved hemodynamics by decreased levels of plasma renin suggesting that these mice have decreased symptoms of shock compared to Il22^{-/-} mice alone. However, the authors should also include data to demonstrate the changes in immune cell composition of Il22^{-/-} mice and Il22^{-/-} mice given fluid resuscitation as Il22^{-/-} mice have been shown to have decreased numbers of neutrophils and other immune cells that contribute to Cr clearance. This is especially relevant since the authors show no differences in bacterial burden in Fig. 3I and Fig. 4A-B in Il22^{-/-} mice or Il22^{-/-} mice given fluid resuscitation. The authors should address if the fluid resuscitation maintains hemodynamic stability long enough for compensatory immune mechanisms to clear the pathogen at later time points. In addition, the authors should include quantitative histological data that scores inflammation and epithelial cell damage, as well as show bacterial attachment to distal colonocytes at the peak of infection.

Major Point 2: Downregulation of Solute Carriers

In Figure 2, the authors demonstrate that several solute carriers are downregulated in infected Il22^{-/-} mice compared to infected controls. Many of these solute carriers, and carbonic anhydrases like Ca1, have also been shown to be downregulated in Cr-infected WT mice. The authors should address how the significant pathology caused by infection during the early immune response leads to epithelial cell damage and increased epithelial cell turnover, potentially altering the cellular sources of these solute carriers even in WT mice. Furthermore, the authors should compare the expression

levels of these carriers and enzymes in naïve WT and Il22^{-/-} mice to better contextualize their findings. In addition, changes in IL-22-inducible genes should be shown as positive controls in Figures 2 and 4 for a clearer comparison between naïve and infected mice.

Major Point 3: Bacterial Load and Attachment in Different Groups

In Figure 3, the authors present significant differences in fecal CFU between infected Il22^{-/-} and Il22^{-/-} + FT mice. It would be helpful to include data on the quantity and localization of bacterial attachment to the epithelium in infected WT, Il22^{-/-} and Il22^{-/-} + FT mice to fully assess the impact of treatment.

Major Point 4: E-Cadherin and Histological Analysis

In Figure 4C, the E-cadherin stain in the infected mice appears identical to the DAPI stain, with similar nuclear staining in LP cells. The current images shown should be replaced with those showing E-cadherin-stained tissue. In addition, as the images are not cross-sections and are taken with different objectives (resulting in different scale bars), it is difficult to assess difference in crypt length in naïve and infected mice. Quantitative histology scoring should be incorporated into this figure. Moreover, including images showing bacterial localization in the different experimental groups would enhance the manuscript. It is important to determine whether Cr colonizes colonic crypts in infected Il22^{-/-} and Il22^{-/-} + FT mice compared to controls.

Major Point 5: Inconsistent Quantification of Colon Length and Histological Data

In Figure 5, there are inconsistencies between the quantitation of colon length (Fig. 5A) and the corresponding colon images (Fig. 5B), especially when comparing day 0 and day 9. In addition, Fig. 5B is inconsistent with the immunostaining shown in Figure 4C and histology in Fig. 5D. Quantitative histological scoring is needed for a more accurate comparison of the experimental groups. The colon from WT Cr day 9 infected mice (Fig. 5B) shows normal stool, which aligns with Fig. 1C and further suggesting a less severe phenotype in WT mice compared to historical studies.

Minor Comments:

Minor point 1:

The color scheme for WT, KO and KO+ FT groups should be consistent across Figure 3. In addition, the legends for Figs. 3B and C use the same labels but display different colors and symbols on the graph. This inconsistency is also seen in Figs E and F.

Minor point 2:

In Figure 4C, the scale bars in the uninfected versus infected samples should be identical to facilitate comparison of groups.

Minor point 3:

Figure 4D would be more informative if some of the genes were labeled next to the heatmap and the DEGs listed in a table or excel spreadsheet in the supplemental data section.

Reviewer #2

(Remarks to the Author)

Citrobacter rodentium causes a lethal infection in IL-22 ko mice, a phenotype that has been highly reproducible by many research groups. Yet, the precise mechanism by which these mice succumb to infection has remained elusive. By performing proteomics, Mishra et al identified a downregulation in colonic ion transporters and a consequent electrolyte imbalance. Fluid therapy restored electrolyte homeostasis and mouse survival, without affecting bacterial colonization, dissemination, or epithelial integrity. Compensatory mechanisms (likely mediated by IL-17F and/or IL-10) mediated epithelial regeneration and protection against reinfection.

Overall, the manuscript addresses a significant problem, the experiments are well conducted, and the results are convincing. My main issue is that the authors claim in several places that the observed phenotypes are not due to IL-22 and that IL-22 is indispensable for survival. If these electrolyte transport genes are downregulated in mice lacking IL-22, this likely means they are also essential components of the IL-22-induced responses. No results in the manuscript suggest that IL-22 plays no role in survival.

1) The authors should rephrase/ tone down their statements about the findings not being related to IL-22 (clearly, they are seen in mice that lack IL-22). See examples below:

Ln 27-28: "These findings overturn the long-standing paradigm that IL-22 is indispensable for host survival from CR infection."

Ln 88-90: "Furthermore, we show that fluid therapy (FT) alone is sufficient to rescue Il22^{-/-} mice, challenging the prevailing notion that IL-22 is indispensable for survival following CR infection."

Ln 204-205: "These findings challenge the prevailing dogma regarding the indispensability of IL-22 during CR infection."

Ln 239-240: "These findings suggest that while IL-22 contributes to epithelial maintenance during infection, it is dispensable and can be compensated by IL-22-independent mechanisms."

2) What happen if IL-22^{-/-} mice are treated with recombinant IL-22? Does IL-22 treatment rescue the electrolyte imbalance? What about in vitro, in colonocyte models? Does IL-22 induce expression of these ion transporters?

3) What is the expression of other IL-22-induced genes (e.g., Reg3b, Reg3g) in IL-22^{-/-} mice treated with FT, in comparison to IL22^{+/+} and IL-22^{-/-} untreated? Does it change over time?

Reviewer #3

(Remarks to the Author)

The paper by Mishra et al delves into the molecular mechanisms of mortality by *Citrobacter rodentium*. The central premise of the paper that IL22 is not the main protective driver, but dehydration is exceedingly well supported with the double knock-out mice that can be rescued with simple fluid replacement. I can go along with the author's conclusion that this can inform clinical practice on 'therapeutic interventions that complement host defence mechanisms'.

The paper is extremely well written. I do have questions about the data analysis with regards to the proteomics part. I don't think this will impact the conclusions to far ranging, but I do think they need to be taken into consideration. As such, I would recommend acceptance with major revisions.

Minor

[1] Not a breaking point, but I find MS2 collection in the ion trap for peptide identification odd. We know that the low resolution from this device leads to misinterpretation.

[2] In sample preparation for TMT labelling, can the authors add the full buffer details the peptides were solubilized in prior to labelling?

[3] "These findings suggest 163 that FT does not directly modulate the epithelial proteome but instead functions as a 164 supportive measure to counteract dehydration.", I do not understand this conclusion. I guess that colon alone were measured (detail that still needs to be added to the methods), which if memory serves does consist of other cells than solely epithelial cells. The authors could therefore be blind to compensatory effects from other cells. And then there is PTM regulation (phosphorylation for example) that is completely ignored!

Major

[1] Figure 2 is a bit of a disaster. For example, the legend for 'No. proteins' looks like it belongs to panel A but in fact belongs to panel B. The volcano plot in panel A is massively over-representing the changes in the data with completely arbitrary cutoff values. For example, the p-value is cut at 0.05 and I would estimate the minimal fold-change at 1.4-fold. There are good statistical approaches to calculating these cutoff values based on the data (for example, as implemented in the Perseus analysis software from the Jurgen Cox lab). I would estimate that a lot of significant hits would disappear (based on the errorbars, Ca1 – especially as these errorbars I think are calculated from the SEM, which is fine to use but the bars are excessive in that light). For panel B, the GO enrichment is typically not very precise. I would suggest discarding this panel (or move to the supplementary) in favour of only highlighting the list of proteins associated with/that are ion channels like presented in presented in panel C.

[2] Figure 3H, unclear to me whether the loss of weight is attributable to loss of water, which would be the parameter of interest to support the conclusions of the paper. Is there a measure that can be used to quantify this? I know the scales at my local gym do this, but unsure how accurate that is.

[3] Figure 4C, I'm having trouble to see from the stainings whether there is any loss of E-cadherin expression in the FT treated? Likewise, for figure 5D I really have trouble to read the weight to length ratio? Very confusing both, would be good to quantify this somehow.

[4] While I find the comet search engine for proteomics data fantastic as it is open source and accessible, I would not call this the most reliable option out there. I think the data needs to be reprocessed using a more accepted pipeline like MaxQuant or MSFragger. Both are freely available as well, so there is no reason not to use them.

[5] Missing value imputation for the TMT channels is done against the lowest intensity value. This will produce artefacts in the statistical analysis later on. Imputation is a fact of life, but at the very least the authors should select from a normal distribution at the low end of the detectable intensity scale. Also here, Perseus has a good implementation (I would like to point out that I am not involved with the development of that software).

[6] Perseus was used for ANOVA. Can the authors provide details on the FDR correction that was applied? Perseus does have more options than simply to cutoff at $p < 0.05$, which statistically speaking doesn't make any sense.

[7] Figure 5E. I find it somewhat of a stretch to rely solely on a PCA to make the statement that the proteome reverts to normal. Yes, I agree it does seem to somewhat indicate this (although also here, one is blind to compensatory effects). It would be much more insightful to look at the protein behaviors. Boxplots seem to me a good way of showcasing how things revert.

Version 1:

Reviewer comments:

Reviewer #1

(Remarks to the Author)
Response to Revisions (Manuscript #NCOMMS-25-15027)

Mishra et. al.: "Rehydration rescues Il22^{-/-} mice from lethal *Citrobacter rodentium* infection"

While the authors have made substantial improvements following my initial feedback, a fundamental mechanistic contradiction remains unresolved, and critical methodological issues persist. These concerns significantly impact the study's interpretation and clinical relevance.

Improvements to the Manuscript:

Mishra and colleagues have significantly improved their manuscript by including several important additions:

- Additional biological replicates have been incorporated;
- Naïve controls were added to strengthen the experimental design;
- Histology scoring and representative images were expanded;
- Diarrhea in WT mice was confirmed, addressing concerns about variability in infection severity;
- Discrepancies in infection severity were compared to published studies, although some variation may still exist due to strain differences in *Citrobacter rodentium* or variations in bacterial preparation;
- New findings were added showing that fluid therapy reduces blood neutrophil counts and is associated with increased tissue expression of IL-17F and IL-10, suggesting a possible immunomodulatory role.

These additions substantially improve the rigor of the study and contribute novel observations suggesting that fluid therapy may influence immune responses during intestinal infection. However, the abstract and conclusions have not been adequately revised to incorporate new mechanistic insights regarding cytokine changes or acknowledge the limitations in understanding how fluid therapy overcomes transporter defects. This represents a missed opportunity to present a more nuanced interpretation of the findings.

The Central Mechanistic Problem:

The most critical unresolved issue is a fundamental contradiction in the proposed mechanism. If solute carriers responsible for ion and water transport are severely downregulated in infected Il22^{-/-} mice, how does systemic fluid therapy restore intestinal hydration? The authors acknowledge this limitation but provide no experimental evidence to resolve it. If epithelial transporters are non-functional, systemic fluid administration should not enhance mucosal hydration through physiological pathways. This mechanistic disconnect undermines the study's central thesis and requires direct functional validation to determine whether fluid therapy affects transporter function, paracellular permeability, or alternative absorptive pathways. Clarifying this point would significantly strengthen the mechanistic interpretation and highlight the novelty of fluid therapy as more than just a supportive measure.

Statistical and Experimental Rigor:

The proteomics analysis remains fundamentally flawed. Despite using only two biological replicates, the authors deliberately chose not to apply false discovery rate correction because, as they acknowledge, "when applying multiple hypothesis correction methods such as Benjamini-Hochberg FDR adjustment, all p-values exceed the significance threshold due to the low sample size". This admission reveals that their findings would not survive appropriate statistical correction, fundamentally undermining their proteomics conclusions. The study also lacks proper experimental controls including sham injection groups, comparison with oral rehydration therapy, and assessment of different fluid compositions or dosing regimens.

Clinical Translation:

The clinical implications remain overstated despite reviewer feedback. Human infectious diarrhea typically involves multiple pathogens with different pathophysiology from this single-pathogen mouse model. Most clinical diarrhea is managed with oral rehydration, not the intensive subcutaneous fluid therapy demonstrated here. The resource-intensive intervention may not be practical where infectious diarrhea is most prevalent.

Summary and Recommendations:

While the authors have made commendable efforts to address reviewer concerns, the manuscript suffers from a fundamental mechanistic gap that undermines its central conclusions. The core question of how systemic fluid therapy can restore epithelial function when the molecular machinery for ion and water transport is severely impaired remains unanswered.

The authors should prioritize addressing the mechanistic contradiction through direct functional measurements using techniques such as Ussing chamber analysis. They must include proper experimental controls to distinguish specific therapeutic effects from non-specific hemodynamic or stress-response effects. Most importantly, they need to provide mechanistic evidence for how fluid therapy overcomes transporter defects or acknowledge that the intervention may work through entirely different pathways than proposed. The limitations of the study should be acknowledged more explicitly in the abstract and conclusions and claims about clinical relevance and therapeutic implications should be substantially moderated.

Reviewer #2

(Remarks to the Author)

The authors have addressed my previous comments, performed new experiments to further support the study's conclusions, and revised the text based on my prior suggestions. I have no further comments and I commend the authors on this important work.

Reviewer #3

(Remarks to the Author)

All concerns successfully dealt with

We thank the reviewers for their feedback and suggestions, which were fully addressed experimentally and analytically. This resulted in increasing the robustness and clarity of the manuscript.

In line with the reviewers' recommendations, we have avoided the term "IL-22 dispensability" and instead emphasised on its role in preventing fatal dehydration during CR infection.

For clarity, Reviewer#1 major point #1, #2, #5 have been subdivided into separate paragraphs and addressed individually.

Reviewer #1

Major Point 1: Disease Severity Discrepancy

The severity of phenotypes in both WT and *Il22*^{-/-} mice after infection appears less pronounced than historical data suggests. *Il22*^{-/-} mice begin to succumb to infection by day 7 in several reported studies using the Cr model (Zheng et al., 2008, Nature Medicine; Zindl et al., 2022, Immunity), whereas in this study *Il22*^{-/-} mice do not succumb to infection until day 10. This difference may be attributed to variations in the *C. rodentium* strain used, experimental protocol, or differences in gut microbiota composition. Additionally, within this manuscript there are major differences in severity as shown when comparing *Il22*^{-/-} mice survival data in Fig. 1B to *Il22*^{-/-} mice survival data in Fig. 3G. In Fig. 1B, mice survive until day ~14 whereas all mice succumb by day 10 in Fig. 3G.

Response: In the original version of the manuscript, Figures 1B and 3G each presented data from single experiments, as was stated in the figure legends. We have now replaced Fig. 1A-B with pooled survival data from five independent experiments performed with WT and *Il22*^{-/-} mice. Likewise, the revised Fig. 3H-I (previously 3G-H) now show pooled data from three independent experiments with WT and *Il22*^{-/-} mice, with or without FT.

To provide broader context, we have compiled survival data from major published studies (Table S1). Our results align closely with these reports. Notably, we consistently observe 100% mortality in *Il22*^{-/-} mice in our colony.

Furthermore, the mild diarrhea in WT mice and the delayed mortality in *Il22*^{-/-} mice could indicate an underlying SFB infection or another pathobiont, potentially leading to milder symptoms and delayed peak of bacterial burden (Fig. 1B-D).

Response: The above listed new figures show that our data are aligned with previous reports and that our mice do not show milder symptoms or delayed peak of bacterial burden. Moreover, as could be expected, our WT and *Il22*^{-/-} mice are bred and maintained in a clean-air facility at Imperial College London under SPF conditions, as described in the Methods. Comprehensive health screening is conducted every three months, including microbiological testing of oral swabs, faecal samples, pelt swabs, and serum. Reports are available upon request.

Fig. 1C shows no diarrhea in *Il22*^{+/+} mice when it is widely reported that wild-type mice also develop clinical symptoms of acute Cr infection. The attenuated phenotype seen in these mice is a potential confounding variable in assessing the relevance of fluid resuscitation for rescuing *Il22*^{-/-} mice and the claim by the authors that IL-22 is dispensable in the Cr model.

Response: We have added statistical analysis to Fig. 1 demonstrating a significant increase in faecal water content in WT mice post-infection. Additionally, we have now included faecal trait scores using the Bristol stool scoring method for laboratory mice to quantitatively assess stool consistency (new Fig. 1D). As expected, WT mice exhibit clear signs of diarrhoea; however, C57BL/6 mice do not develop diarrhoea as severe as that observed in susceptible strains such as C3H, FVB, or *Il22*^{-/-} mice. Over the past three decades, our group has studied CR and a plethora of CR mutant infections in multiple mouse strains (e.g. Biswas *et al.*, *Gut Microbes*, 2025, PMID: 40599135), and the WT phenotype reported here is consistent with established outcomes for C57BL/6 mice.

Response to reviewers' comments NComms-25-15027

In addition, the colon pathology, including hyperplasia, appears similar between infected WT and *Il22*^{-/-} mice (Figs. 4C and 5D).

Response: Yes, the observation is correct and quantification of hyperplasia at 9 dpi revealed no significant difference between WT and *Il22*^{-/-} mice (new Fig. 4K). However, *Il22*^{-/-} mice displayed greater epithelial damage and inflammation (Fig. 4I-J). PCNA staining indicated a larger proliferative zone in *Il22*^{-/-} mice (new Fig. 4L, Fig. S3), and at 20 dpi they exhibited significantly greater hyperplasia (new Fig. 7F). This suggests that similar hyperplasia values at 9 dpi may underestimate the proliferative response in *Il22*^{-/-} mice due to increased epithelial abrasion. Supporting this, hyperplasia measurements at 6 dpi shown below—when epithelial damage is less severe—show significantly higher values in *Il22*^{-/-} mice.

It is conceivable that fluid resuscitation would improve the hemodynamic parameters of *Il22*^{-/-} mice that have overwhelming bacterial burden, as has been previously reported to occur in *Il22*^{-/-} mice. The authors here suggest improved hemodynamics by decreased levels of plasma renin suggesting that these mice have decreased symptoms of shock compared to *Il22*^{-/-} mice alone. However, the authors should also include data to demonstrate the changes in immune cell composition of *Il22*^{-/-} mice and *Il22*^{-/-} mice given fluid resuscitation as *Il22*^{-/-} mice have been shown to have decreased numbers of neutrophils and other immune cells that contribute to Cr clearance. This is especially relevant since the authors show no differences in bacterial burden in Fig. 3I and Fig. 4A-B in *Il22*^{-/-} mice or *Il22*^{-/-} mice given fluid resuscitation. The authors should address if the fluid resuscitation maintains hemodynamic stability long enough for compensatory immune mechanisms to clear the pathogen at later time points.

Response: We have now included complete blood count (CBC) data for WT and *Il22*^{-/-} mice before and after infection (new Fig. 5A-E). Both genotypes displayed leukopenia, lymphopenia, and thrombocytopenia (which as far as we know have never been reported in this context). Uniquely, *Il22*^{-/-} mice exhibited peripheral neutrophilia, which was significantly reduced by FT. Although *Il22*^{-/-} mice have been reported to express lower levels of neutrophil-attractant chemokines (e.g., *Cxcl1*, *Cxcl2*, *Cxcl5*) in colonic IECs (Zindl *et al.*, *Immunity*, 2022, PMID: 35263568), histological studies have documented higher number of colonic neutrophils (Melchior *et al.*, *Infect Immun*, 2024, PMID: 38557196). Consistent with this, we observed increased neutrophil infiltration in the submucosa of *Il22*^{-/-} mice, with no difference following FT (new Fig. 5F, Fig. S3, Fig. S5J). Consistent with the established role of IL-22 in recruiting neutrophils to the colonic epithelium in mice (Zindl *et al.*, *Immunity*, 2022, PMID: 35263568) and in patients with colitis (Pavlidis *et al.*, *Nat. Commun*, 2022, PMID: 36192482), *Il22*^{-/-} mice exhibited a lower proportion of mucosal neutrophils relative to the total colonic neutrophil population (new Fig. S3B).

In addition, the authors should include quantitative histological data that scores inflammation and epithelial cell damage, as well as show bacterial attachment to distal colonocytes at the peak of infection.

Response: This has now been added (new Fig. 4B, 4J).

Major Point 2: Downregulation of Solute Carriers

In Figure 2, the authors demonstrate that several solute carriers are downregulated in infected *Il22^{-/-}* mice compared to infected controls. Many of these solute carriers, and carbonic anhydrases like Ca1, have also been shown to be downregulated in CR-infected WT mice. The authors should address how the significant pathology caused by infection during the early immune response leads to epithelial cell damage and increased epithelial cell turnover, potentially altering the cellular sources of these solute carriers even in WT mice.

Response: We have now expanded our analysis to fully address this point. First, we included qRT-PCR data comparing uninfected and infected WT and *Il22^{-/-}* mice for solute carriers (new Fig. 2B-C), which confirmed that expression of these transporters is further reduced in CR infected *Il22^{-/-}* mice.

To determine the cellular origin of these solute carriers, we analysed publicly available scRNA-seq data from mouse colon (Single Cell Portal accession ID SCP1891) (new Fig. 2D). This revealed that the selected ion transporters are almost exclusively expressed by absorptive enterocytes. We next analysed colonic epithelial subpopulations using qRT-PCR analysis for marker genes (new Fig. 2E). This analysis revealed a significantly greater loss of absorptive enterocyte markers in CR infected *Il22^{-/-}* mice (new Fig. 2E).

Furthermore, to directly test whether IL-22 regulates ion transporter expression, we treated naïve mice with recombinant IL-22 and assessed transporter expression in the colon (new Figs. S2B-C). This demonstrated that IL-22 stimulation does not directly induce expression of the selected ion channels. (Please also refer response to Reviewer#2, Major point 2 for detailed approach).

These results together suggest that reduced solute carrier expression in *Il22^{-/-}* mice is not due to direct regulation by IL-22. Instead, it reflects the greater loss of absorptive enterocytes during infection.

Furthermore, the authors should compare the expression levels of these carriers and enzymes in naïve WT and *Il22^{-/-}* mice to better contextualize their findings.

Response: This has now been added (new Fig. 2A-C)

In addition, changes in IL-22-inducible genes should be shown as positive controls in Figures 2 and 4 for a clearer comparison between naïve and infected mice.

Response: We have now included the expression profiles of established IL-22-inducible genes (*Reg3b*, *Reg3g*) as positive controls (new Fig. 2B). In addition, we have quantitative data of the levels of IL-22 in the colonic explant culture, where no detectable IL-22 was observed in *Il22^{-/-}* mice (new Fig. 6A, 7G). We also performed recombinant IL-22 treatment to CR-infected *Il22^{-/-}* mice which rescued the mice and resulted in pathology similar to WT mice (please see response to Reviewer#2 major comment#2).

Major Point 3: Bacterial Load and Attachment in Different Groups

In Figure 3, the authors present significant differences in fecal CFU between infected *Il22^{-/-}* and *Il22^{-/-}* + FT mice. It would be helpful to include data on the quantity and localization of bacterial attachment to the epithelium in infected WT, *Il22^{-/-}* and *Il22^{-/-}* + FT mice to fully assess the impact of treatment.

Response: We have now included CR staining to show the localisation of bacterial attachment to the distal colon in WT, *Il22^{-/-}*, and *Il22^{-/-}* + FT mice at the peak of infection (new Fig. 4B).

Major Point 4: E-Cadherin and Histological Analysis

In Figure 4C, the E-cadherin stain in the infected mice appears identical to the DAPI stain, with similar nuclear staining in LP cells. The current images shown should be replaced with those showing E-cadherin-stained tissue. In addition, as the images are not cross-sections and are taken with different objectives (resulting in different scale bars), it is difficult to assess

difference in crypt length in naïve and infected mice. Quantitative histology scoring should be incorporated into this figure. Moreover, including images showing bacterial localization in the different experimental groups would enhance the manuscript. It is important to determine whether Cr colonizes colonic crypts in infected *Il22^{-/-}* and *Il22^{-/-}* + FT mice compared to controls.

Response: We have replaced the original E-cadherin images (new Fig. 5F, Fig. S3). Figure S3 illustrates the distinction between E-cadherin staining, which outlines the cell periphery, and DAPI staining, which labels nuclei.

Additionally, we have performed an intestinal permeability assay using the FITC–dextran to complement the E-cadherin staining (new Fig. 4D). *Il22^{-/-}* mice, with or without FT, exhibited higher serum FITC–dextran levels than WT mice.

We have incorporated quantitative histological scoring for inflammation and epithelial cell damage at 9 dpi (new Fig. 4J), as well as at 20 and 48 dpi (new Fig. 7E). Also, see response to Major point 3 for CR localisation comment.

Major Point 5: Inconsistent Quantification of Colon Length and Histological Data

In Figure 5, there are inconsistencies between the quantitation of colon length (Fig. 5A) and the corresponding colon images (Fig. 5B), especially when comparing day 0 and day 9.

Response: We have revised Fig. 4G to include representative colon images selected to closely match the mean colon length values shown in the corresponding graph. In Fig. 7C, we have retained the original colon images to illustrate the full extent of disease severity in *Il22^{-/-}* mice and their subsequent recovery; this is now stated in the figure legend.

In addition, Fig. 5B is inconsistent with the immunostaining shown in Figure 4C and histology in Fig. 5D. Quantitative histological scoring is needed for a more accurate comparison of the experimental groups. The colon from WT Cr day 9 infected mice (Fig. 5B) shows normal stool, which aligns with Fig. 1C and further suggesting a less severe phenotype in WT mice compared to historical studies.

Response: New immunostaining and histological scoring have been added (also see response to Major point 1 and 4).

We would like to reemphasise that the CR-mediated disease reported in this manuscript is not attenuated in either WT or *Il22^{-/-}* mice. Data from uninfected controls have now been included **for all disease parameters**, with statistical comparisons between infected and uninfected groups. As expected for CR infection in WT mice, we observed increased faecal water content (Fig. 1E), upregulation of AMPs (new Fig. 2B), downregulation of enterocyte markers (*Slc26a3*, *Ca4*, *Alpi*, *Krt20*) (new Fig. 2C), reduced colon length (Fig. 4F), colonic hyperplasia (new Fig. 4K), increased colonic immune cell infiltration (new Fig. S5), elevated pro-inflammatory cytokine levels (new Fig. 6), and persistent IL-22 and IFN γ levels after recovery (new Fig. 7G-H). In *Il22^{-/-}* mice, infection was severe, with 100% mortality observed, and disease parameters are reported throughout the manuscript.

Minor Comments:

Minor point 1:

The color scheme for WT, KO and KO+ FT groups should be consistent across Figure 3. In addition, the legends for Figs. 3B and C use the same labels but display different colors and symbols on the graph. This inconsistency is also seen in Figs E and F.

Response: This has been corrected.

Minor point 2:

In Figure 4C, the scale bars in the uninfected versus infected samples should be identical to facilitate comparison of groups.

Response: This has been addressed.

Minor point 3:

Figure 4D would be more informative if some of the genes were labeled next to the heatmap and the DEGs listed in a table or excel spreadsheet in the supplemental data section.

Response: An excel sheet with all the proteins and expression value are provided in the source data file. Private link to the data: <https://figshare.com/s/ec3ea0f050eedde60d8f>

Reviewer #2

Major comment 1: Rephrasing IL-22 dispensability

The authors should rephrase/tone down their statements about the findings not being related to IL-22 (clearly, they are seen in mice that lack IL-22). See examples below:

Ln 27-28: "These findings overturn the long-standing paradigm that IL-22 is indispensable for host survival from CR infection."

Ln 88-90: "Furthermore, we show that fluid therapy (FT) alone is sufficient to rescue *Il22*^{-/-} mice, challenging the prevailing notion that IL-22 is indispensable for survival following CR infection."

Ln 204-205: "These findings challenge the prevailing dogma regarding the indispensability of IL-22 during CR infection."

Ln 239-240: "These findings suggest that while IL-22 contributes to epithelial maintenance during infection, it is dispensable and can be compensated by IL-22-independent mechanisms."

Response: To accommodate this useful comment, we have revised the manuscript by removing statements implying that IL-22 is dispensable during CR infection. Instead, we now emphasise the role IL-22 plays in preventing fatal dehydration during infection. Our findings therefore refine the understanding of IL-22 biology by showing that survival of *Il22*^{-/-} mice can be achieved when dehydration is corrected, even though IL-22 remains integral to epithelial protection and host physiology.

Major comment 2: *Il22*^{-/-} mice complementation

What happen if *Il22*^{-/-} mice are treated with recombinant IL-22? Does IL-22 treatment rescue the electrolyte imbalance? What about in vitro, in colonocyte models? Does IL-22 induce expression of these ion transporters?

Response: We treated CR-infected *Il22*^{-/-} mice with 2 µg of recombinant IL-22 (rIL-22), intraperitoneally from 4 dpi onwards. As reported previously (Basu et al., *Immunity*, 2012, PMID: 23200827), rIL-22 treatment fully rescued CR-infected *Il22*^{-/-} mice, which presented a similar pathology to WT mice (shown below). In contrast, untreated *Il22*^{-/-} mice lost weight, developed diarrhoea, and succumbed to infection, as described in this study.

Therefore, to directly address whether IL-22 regulates the expression of ion transporters, we firstly analysed publicly available transcriptomic data from mouse colonoids treated with recombinant IL-22 (new Fig. S2A). In addition, we also treated naïve mice with rIL-22 and assessed the expression of these ion channels in the colon (new Figs. S2B-C). These analyses provide both in vitro and in vivo evidence that IL-22 does not directly regulate the levels of these ion channels. Instead, the further reduction in their expression in *Il22*^{-/-} mice is attributable to the greater loss of absorptive enterocytes (also see our response to Reviewer #1, Major Point 1).

Major comment 3: IL-22-induced genes

What is the expression of other IL-22-induced genes (e.g., *Reg3b*, *Reg3g*) in *Il22*^{-/-} mice treated with FT, in comparison to *Il22*^{+/+} and *Il22*^{-/-} untreated? Does it change over time?

Response: We have now included the expression profiles of *Reg3b* and *Reg3g* in *Il22*^{-/-} mice treated with FT, compared with untreated *Il22*^{-/-} controls (new Fig. S6). FT-treated mice displayed significantly higher *Reg3g* levels in the colon.

In addition, we replotted the *Reg3b* and *Reg3g* expression data from Fig. 2 and Fig. S6 to illustrate their expression dynamics over the course of CR infection in *Il22*^{-/-} mice (shown below). Although expression of these IL-22-inducible AMPs remained significantly lower than in WT mice (new Fig2B), both genes were markedly upregulated following infection, indicating IL-22-independent regulation upon CR infection.

Reviewer #3

Minor

1) Not a breaking point, but I find MS2 collection in the ion trap for peptide identification odd. We know that the low resolution from this device leads to misinterpretation.

Response: While we acknowledge the inherent trade-off between resolution and scan speed, the use of ion trap MS2 in combination with synchronous precursor selection (SPS)-MS3 quantification is a well-established and validated strategy for TMT-based proteomics. This approach was originally developed and optimized by the Gygi laboratory to enhance quantitative accuracy by reducing ratio distortion due to co-isolation interference, while maintaining proteome depth through high-throughput MS2 scans in the ion trap (Erickson *et al.*, *J Proteome Res*, 2019, PMID: 30658528). Importantly, peptide spectral matches (PSMs) were filtered using target-decoy database searches to control the false discovery rate (FDR) at both the peptide and protein levels, ensuring robust and reproducible identification despite the lower resolution of ion trap MS2.

2) In sample preparation for TMT labelling, can the authors add the full buffer details the peptides were solubilized in prior to labelling?

Response: This has been added to the methods section.

3) "These findings suggest 163 that FT does not directly modulate the epithelial proteome but instead functions as a 164 supportive measure to counteract dehydration.", I do not understand this conclusion. I guess that colon alone were measured (detail that still needs to be added to the methods), which if memory serves does consist of other cells than solely epithelial cells. The authors could therefore be blind to compensatory effects from other cells. And then there is PTM regulation (phosphorylation for example) that is completely ignored!

Response: We apologise for the confusion. The original statement was intended to refer to all parameters measured in the figure and not solely to the epithelial proteome. We have now revised the text accordingly to clarify this point and avoid confusion. **Line 202-203.**

Furthermore, to address potential compensatory changes, we performed FACS analysis of immune cell populations in the colon (new Fig. S5), immunostaining for neutrophils in the colon (new Fig. 5F), and cytokine analysis from colonic tissue (new Fig. 6). FT did not alter these parameters, consistent with our conclusion that FT provide supportive care, primarily counteracts dehydration rather than directly modulating epithelial cell responses. Phosphoproteomics could provide further mechanistic insight, however, this is outside the scope of the present study and does not affect the take home message regarding IL-22 dispensability. Of course, this is part of our future plans, which feature in our recent grant application.

Major

1) Figure 2 is a bit of a disaster. For example, the legend for 'No. proteins' looks like it belongs to panel A but in fact belongs to panel B. The volcano plot in panel A is massively over-representing the changes in the data with completely arbitrary cutoff values. For example, the p-value is cut at 0.05 and I would estimate the minimal fold-change at 1.4-fold. There are good statistical approaches to calculating these cutoff values based on the data (for example, as implemented in the Perseus analysis software from the Jurgen Cox lab). I would estimate that a lot of significant hits would disappear (based on the errorbars, Ca1 – especially as these errorbars I think are calculated from the SEM, which is fine to use but the bars are excessive in that light). For panel B, the GO enrichment is typically not very precise. I would suggest discarding this panel (or move to the supplementary) in favour of only highlighting the list of proteins associated with/that are ion channels like presented in presented in panel C.

Response: We appreciate the reviewer's detailed evaluation of Fig. 2. In response, we have replaced the figure with a more focused presentation that includes a robust heatmap of ion channels and qRT-PCR validation of their downregulation upon infection. As suggested, the revised volcano plot and GO enrichment analysis have been moved to the supplementary data. (also see our response to Reviewer #1, Major Point 2).

We would also like to clarify the rationale behind our approach and the intended interpretation of the proteomics data. The proteomics analysis was designed as a screening experiment to explore broad differences in the colonic epithelial response to CR infection between WT and *Il22^{-/-}* mice subsequent to the observation that the later develop severe diarrhoea (Fig. 1). We have now made appropriate changes to the text to highlighting the rationale. **Line 111.**

The analysis was conducted from two independent experiments, with three or more mice per condition per experiment. Colonic IECs from each experiment were pooled and processed for proteomics as biological duplicates, which we acknowledge limits statistical power.

To account for this, we assessed the global reproducibility of the dataset. The average coefficient of variation (CV) across all quantified proteins was approximately 29%, with a median CV of 21%, consistent with expectations for TMT-labelled global proteomics in

complex tissues. To improve robustness in differential analysis despite low replicates, we applied a fold-change threshold equivalent to at least three times the median CV, specifically, a \log_2 fold-change threshold of ± 0.7 , paired with a nominal p-value threshold of 0.05. While we agree that these cutoffs are conservative and do not account for multiple hypothesis testing, they are justified in the context of a screen designed to generate hypotheses and highlight functionally relevant categories, rather than to establish definitive differential expression.

2) Figure 3H, unclear to me whether the loss of weight is attributable to loss of water, which would be the parameter of interest to support the conclusions of the paper. Is there a measure that can be used to quantify this? I know the scales at my local gym do this, but unsure how accurate that is.

Response: The weight loss observed in $Il22^{-/-}$ mice reflects the overall severity of disease and is likely due to a combination of factors, including fluid loss, anorexia, and adipisia. While quantitative magnetic resonance (QMR) analysis can be used to determine total body water in live animals, this technology is not available in our facility. Nevertheless, our data on serum dehydration parameters, faecal water content, and clinical scoring collectively indicate that dehydration is a major contributor to weight loss in this model.

3) Figure 4C, I'm having trouble to see from the stainings whether there is any loss of E-cadherin expression in the FT treated? Likewise, for figure 5D I really have trouble to read the weight to length ratio? Very confusing both, would be good to quantify this somehow.

Response: We have replaced the original E-cadherin staining with new images (new Fig. 5F) to improve clarity (also see response to Reviewer#1, Major point 4). To make the dataset easier to interpret, the original Fig. 5 has been reorganised so that parameters from the acute phase at 9 dpi are now shown in Fig. 4, while parameters from the recovery phase are presented in Fig. 7.

4) While I find the comet search engine for proteomics data fantastic as it is open source and accessible, I would not call this the most reliable option out there. I think the data needs to be reprocessed using a more accepted pipeline like MaxQuant or MSFragger. Both are freely available as well, so there is no reason not to use them.

Response: We would like to clarify that Comet was used solely for real-time search (RTS) during data acquisition on the Orbitrap mass spectrometer to enable real-time spectral matching and prioritization of SPS-MS3 triggering. This setup improves acquisition efficiency for TMT-based workflows and does not directly determine the final peptide or protein identifications.

For the offline data analysis, we performed a combined database search using both SequestHT and Comet search engines within Proteome Discoverer 3. Peptide-spectrum matches (PSMs) from both engines were filtered using Percolator with a strict 1% false discovery rate (FDR) at the peptide. This dual-search strategy increases robustness and reliability and is supported by built-in integration within Proteome Discoverer.

We acknowledge that this was not fully described in the original Methods section. The revised manuscript now includes detailed information on the full search strategy, including the use of both SequestHT and Comet for database searching, the FDR filtering parameters, and the decoy database setup.

5) Missing value imputation for the TMT channels is done against the lowest intensity value. This will produce artefacts in the statistical analysis later on. Imputation is a fact of life, but at the very least the authors should select from a normal distribution at the low end of the detectable intensity scale. Also here, Perseus has a good implementation (I would like to point out that I am not involved with the development of that software).

Response: We have reprocessed the dataset using a Perseus Gaussian imputation strategy, wherein missing values were replaced by random values drawn from a normal distribution

centered at the low end of the observed intensity range (mean shifted by -1.8 standard deviations, with a width of 0.3 standard deviations). This method better reflects the assumption that missing values arise from low-abundance peptides falling below the detection limit.

The imputed dataset was used for all subsequent statistical analyses, and the Methods section has been updated accordingly to reflect this change. We believe this approach substantially improves the robustness of our findings.

6) Perseus was used for ANOVA. Can the authors provide details on the FDR correction that was applied? Perseus does have more options than simply to cutoff at $p < 0.05$, which statistically speaking doesn't make any sense.

Response: In our dataset, the proteomics analysis was performed from two independent experiments for each condition (minimum of 3 mice per experiment were used and colonic IECs were pooled), which we fully acknowledge limits the statistical power and precludes robust false discovery rate (FDR) correction in a traditional sense.

Indeed, when applying multiple hypothesis correction methods such as Benjamini-Hochberg FDR adjustment in Perseus, all p-values exceed the significance threshold due to the low sample size. As a result, no features are retained as statistically significant after correction. This outcome is expected and reflects the known limitations of applying ANOVA-based inference with only two replicates per group, where estimation of within-group variance is inherently unreliable.

Given these constraints, we treated this proteomic **screen** as a hypothesis-generating tool and applied a nominal p-value threshold of 0.05 in combination with an empirically determined \log_2 fold-change cutoff of ± 0.7 (corresponding to an over 60% change, which is close to three times the median CV). This approach allowed us to prioritize proteins showing consistent trends and biologically meaningful differences, despite the statistical limitations.

To ensure transparency, we have updated the Methods section to state explicitly that:

- ANOVA was performed using Perseus (version 2.0.11)
- Nominal p-values were used without FDR correction due to replicate limitations
- The results are presented as exploratory and not intended to imply definitive statistical significance

We also note that selected findings from this dataset have been supported by independent validation experiments, lending biological confidence to the observed trends.

7) Figure 5E. I find it somewhat of a stretch to rely solely on a PCA to make the statement that the proteome reverts to normal. Yes, I agree it does seem to somewhat indicate this (although also here, one is blind to compensatory effects). It would be much more insightful to look at the protein behaviors. Boxplots seem to me a good way of showcasing how things revert.

Response: We originally used proteome PCA to provide an overview of recovery from infectious colitis at 20 dpi. In response to the reviewer's suggestion, we have now expanded the analysis to include multiple complementary disease parameters, such as colon histology, serum markers, immune cell profiles, and cytokine levels, to comprehensively assess recovery post infection (new Figs. 7-8).

At the protein level, we specifically examined ion transporters that were dysregulated at 9 dpi and found their expression restored by 20 dpi (new Fig. S8), supporting the conclusion that epithelial function had recovered.

In addition, to integrate systemic and local outcomes, we performed PCA on 13 parameters measured in the study, including colon pathology, serum markers, and blood parameters (Fig. 8C). This analysis revealed that uninfected WT and *I122^{-/-}* mice clustered together but separated upon infection at 9 dpi. By 20 and 48 dpi, however, WT and FT-treated *I122^{-/-}* mice

Response to reviewers' comments NComms-25-15027

clustered together, indicating similar colonic and systemic recovery. This integrated PCA strengthens the proteomics findings and provides a holistic visualisation of disease progression and recovery across infection stages.

Reviewer #1

We thank the reviewer for recognising the substantial improvements we have made in the revision. In response to their further comments, we have now revised both the Abstract and the Discussion, incorporating new mechanistic insights and further reinforcing that the proteomics was designed as a discovery screen with follow-up orthogonal validation. We hope that these additions will be acceptable to address the raised concerns.

Response to major comments:

Comment 1. The Central Mechanistic Problem:

The most critical unresolved issue is a fundamental contradiction in the proposed mechanism. If solute carriers responsible for ion and water transport are severely downregulated in infected *Il22*^{-/-} mice, how does systemic fluid therapy restore intestinal hydration? The authors acknowledge this limitation but provide no experimental evidence to resolve it. If epithelial transporters are non-functional, systemic fluid administration should not enhance mucosal hydration through physiological pathways. This mechanistic disconnect undermines the study's central thesis and requires direct functional validation to determine whether fluid therapy affects transporter function, paracellular permeability, or alternative absorptive pathways. Clarifying this point would significantly strengthen the mechanistic interpretation and highlight the novelty of fluid therapy as more than just a supportive measure.

Response: We would like to stress that we have not proposed that systemic fluid therapy (FT) restores intestinal or mucosal hydration or corrects epithelial transporter defects. Rather, our data demonstrate that FT rescues *Il22*^{-/-} mice by correcting **systemic dehydration** that arises secondary to epithelial damage, diarrhoea and excessive fluid losses in the absence of IL-22. As we clearly emphasised, FT-treated *Il22*^{-/-} mice continue to display similar signs of infectious colitis as untreated *Il22*^{-/-} mice (Fig. 4). To further substantiate this point, we have now added **new Fig. S4**, showing qRT-PCR analysis of colonic ion transporters (*Slc26a3*, *Aqp8*, *Slc5a8*, *Slc15a1*, *Ca2*, *Ca4*) and epithelial-cell-subset marker genes (*Lgr5*, *Dclk1*, *Chga*, *Muc2*, *Krt20*, *Alpi*) in untreated and FT-treated *Il22*^{-/-} mice. This shows no significant differences between the groups, confirming that FT does **not** restore transporter expression or epithelial composition. These results reinforce that the beneficial effect of FT is in maintaining euvolemia and not in modifying the underpinning process leading to excessive fluid losses.

We have now also added a dedicated paragraph in the Discussion (**Line 413-419**) clarifying this mechanistic distinction and outlining future directions aimed at delineating the IL-22-independent recovery mechanisms.

Comment 2. Statistical and Experimental Rigor:

The proteomics analysis remains fundamentally flawed. Despite using only two biological replicates, the authors deliberately chose not to apply false discovery rate correction because, as they acknowledge, "when applying multiple hypothesis correction methods such as Benjamini-Hochberg FDR adjustment, all p-values exceed the significance threshold due to the low sample size". This admission reveals that their findings would not survive appropriate statistical correction, fundamentally undermining their proteomics conclusions. The study also lacks proper experimental controls including sham injection groups, comparison with oral rehydration therapy, and assessment of different fluid compositions or dosing regimens.

Response: In our original submission, we indicated that the proteomics analysis was conducted as a discovery-driven screen using two independent biological replicates per condition, each comprising pooled IECs from a minimum of three mice. In the previous revision, we further clarified this design and explicitly acknowledged in the text (Line 657-658)

that it provides limited statistical power and therefore precludes formal FDR correction. We reiterate that the intent of this analysis was to identify broad pathway-level differences rather than to derive definitive quantitative conclusions.

In response to earlier reviewer feedback (Reviewer #3, major comments 1, 6, 7), we had already revised the text to explicitly describe these statistical limitations and to emphasise orthogonal validation. Specifically, key findings from the proteomics screen were independently confirmed by:

- **qRT-PCR** validation of ion transporter genes (Fig. 2)
- **Flow cytometry** analysis of colonic immune cells (Fig. S6)
- **Cytokine profiling** of colonic explant cultures (Fig. 6)
- **Principal component analysis (PCA)** integrating 13 systemic and colonic parameters (Fig. 8C)

Adding further to above, we have also added **new Fig S4** confirming no changes in the expression of ion transporter genes and epithelial subtype markers in FT treated mice compared to untreated *IL22^{-/-}* mice. Altogether, these independent datasets support and extend the proteomic observations and form the basis of our main conclusions. Importantly, none of the study's interpretations rely solely on the proteomics data and do not influence the core conclusion of the manuscript that "IL-22 is important for maintaining epithelial integrity and limiting infection-induced damage, IL-22-independent pathways can compensate in its absence, enabling bacterial clearance and regeneration of the colonic epithelium when systemic hydration is restored."

Nevertheless, we have added lines to Result section explicitly saying the limitation of proteomics data (**Line 114-117, Line 205-206**). We have also added text to Discussion clearly stating that the proteomics dataset provides exploratory molecular context, with all key biological findings corroborated through independent approaches (**Line 405-410**).

Regarding the reviewer's suggestion to compare oral and subcutaneous fluid therapy, we would like to clarify the rationale for our chosen regimen. The rehydration protocol was designed based on established veterinary and preclinical dehydration guidelines (*Hankenson, F.C. (2013). Critical Care Management for Laboratory Mice and Rats (1st ed.). CRC Press. <https://doi.org/10.1201/b15811>*), and previous studies where FT has rescued susceptible mice strains of C3H and FvB from CR infection (*Borenshtein et al., Infect Immun, 2007, PMID: 17470543*). The rehydration protocol has been described in detail in the Methods section. Depending on the dehydration status, mice received up to 3 mL of balanced electrolyte solution per day, divided into three subcutaneous doses. Such volumes are not recommended to be administered orally in acutely ill mice.

Moreover, oral rehydration is unlikely to be effective in this model, as *IL22^{-/-}* mice exhibit severe colonic epithelial damage and marked downregulation of absorptive ion and water transporters (Fig. 2), which would impair luminal fluid uptake. Subcutaneous administration therefore provides the most physiologically appropriate route to correct **systemic dehydration**, the primary cause of mortality in this model, without confounding by local epithelial dysfunction. We now include this clarification in the Discussion to explain why subcutaneous rehydration was used in this study (**Line 366-371**).

Comment 3. Clinical Translation:

The clinical implications remain overstated despite reviewer feedback. Human infectious diarrhea typically involves multiple pathogens with different pathophysiology from this single-

pathogen mouse model. Most clinical diarrhea is managed with oral rehydration, not the intensive subcutaneous fluid therapy demonstrated here. The resource-intensive intervention may not be practical where infectious diarrhea is most prevalent.

Response: Our study does **not** make any direct claims of clinical translation to therapy. The inclusion of FT in this model was designed solely to emulate the supportive-care context routinely provided to patients experiencing severe dehydration during gastrointestinal infections. Our conclusions are confined to the experimental and mechanistic level, demonstrating that maintaining systemic hydration enables IL-22–independent recovery in mice. We have revised the Abstract and Discussion to make this distinction explicit, clarifying that the translational relevance of our findings lies in refining the design of preclinical models, not in proposing clinical interventions. These clarifications ensure that the scope and applicability of our work are clearly defined.

Summary and Recommendations:

While the authors have made commendable efforts to address reviewer concerns, the manuscript suffers from a fundamental mechanistic gap that undermines its central conclusions. The core question of how systemic fluid therapy can restore epithelial function when the molecular machinery for ion and water transport is severely impaired remains unanswered.

The authors should prioritize addressing the mechanistic contradiction through direct functional measurements using techniques such as Ussing chamber analysis. They must include proper experimental controls to distinguish specific therapeutic effects from non-specific hemodynamic or stress-response effects. Most importantly, they need to provide mechanistic evidence for how fluid therapy overcomes transporter defects or acknowledge that the intervention may work through entirely different pathways than proposed. The limitations of the study should be acknowledged more explicitly in the abstract and conclusions and claims about clinical relevance and therapeutic implications should be substantially moderated.

We have now comprehensively clarified the points raised:

- **Mechanistic clarification:** We explicitly state that FT does **not** restore epithelial hydration or transporter function. Instead, FT corrects systemic dehydration, enabling recovery through IL-22–independent mechanisms. This is supported by the new Fig. S4, which shows no change in ion transporter or epithelial subset gene expression following FT, confirming that the intervention acts through systemic rather than epithelial pathways.
- **Acknowledgement of limitations and future directions:** We have added a new paragraph in the Discussion outlining the limitations of the current study and specifying that future work is required to delineate IL-22 independent mechanisms of CR clearance and epithelial recovery.
- **Statistical and experimental transparency:** We have expanded the Results and Discussion to clarify that the proteomics dataset was exploratory due to limited sample size and that all key findings were validated through independent methodologies (qRT-PCR, cytokine profiling, flow cytometry, and PCA).
- **Moderation of claims:** The Abstract and Discussion have been revised to explicitly limit translational statements to preclinical model refinement, without implying direct human clinical application.

These revisions collectively address all aspects of the reviewer's recommendations. We hope this transparent presentation of limitations and mechanistic framing clarifies the conceptual intent of our work.